# MHPP: Exploring Capabilities and Limitations of Language Models Beyond Basic Code Generation

## Abstract

Recent advancements in large language models (LLMs) have greatly improved code generation, specifically at the function level. For instance, GPT-4o has achieved a 91.0% pass rate on HumanEval. However, this draws into question the adequacy of existing benchmarks in thoroughly assessing function-level code generation capabilities. Our study analyzed two common benchmarks, HumanEval and MBPP, and found that these might not thoroughly evaluate LLMs' code generation capacities due to limitations in quality, difficulty, and granularity. To resolve this, we introduce the Mostly Hard Python Problems (MHPP) dataset, consisting of 210 unique human-curated problems. By focusing on the combination of natural language and code reasoning, MHPP gauges LLMs' abilities to comprehend specifications and restrictions, engage in multi-step reasoning, and apply coding knowledge effectively. Initial evaluations of 26 LLMs using MHPP showed many high-performing models on HumanEval failed to achieve similar success on MHPP. Moreover, MHPP highlighted various previously undiscovered limitations within various LLMs, leading us to believe that it could pave the way for a better understanding of LLMs' capabilities and limitations.

## 1 Introduction

Large language models (LLMs) have recently driven striking performance improvements across various tasks (Ouyang et al., 2022; Touvron et al., 2023; OpenAI, 2023). Recent models such as Llama 3.1 (Dubey et al., 2024), CodeLlama (Rozière et al., 2023), CodeGemma (Team et al., 2024), and GPT-4o (OpenAI, 2024) have been successful in demonstrating their efficacy in code-related tasks from program repair (Haque et al., 2022; Jin et al., 2023) to automated testing (Lemieux et al., 2023; Schäfer et al., 2024). LLMs are utilized to develop innovative tools aimed at aiding programmers to write code more efficiently (Chen et al., 2021).

Code generation is a key area for evaluating LLMs' capabilities. Code generation broadly spans converting natural language prompts into executable code, not limited to predefined templates such as function signatures and docstrings. This process can range from pure text descriptions to complete code generation, emphasizing the versatility and adaptability required for LLMs. Our focus is on Function-Level Code Generation. An example is illustrated in Figure 1. It emphasizes the translation of natural language into functional code, underlining natural language comprehension's importance for creating accurate programming constructs. Benchmarks like HumanEval (Chen et al., 2021) and MBPP (Austin et al., 2021) stand out in assessing these models, showcasing LLMs' strengths in code generation through their understanding of natural language and generation abilities. For instance, GPT-4o (OpenAI, 2024) achieves a 91.0% pass rate on HumanEval (Chen et al., 2021).

However, on existing benchmarks, performance differences between models are insignificant - all achieve high pass rates. We thus raise two concerns: 1) Basic datasets lack discriminative power to distinguish model capabilities, making it difficult to assess their relative strengths and weaknesses. 2) High overall pass rates on existing tasks alone cannot determine if models have truly mastered functional programming competency and encoding skills to address diverse challenges. To answer these questions, we conducted detailed experiments with strong code models on the market, including closed-source models like GPT-4 (OpenAI, 2023), GPT-3.5 (OpenAI, 2022), and open-source models

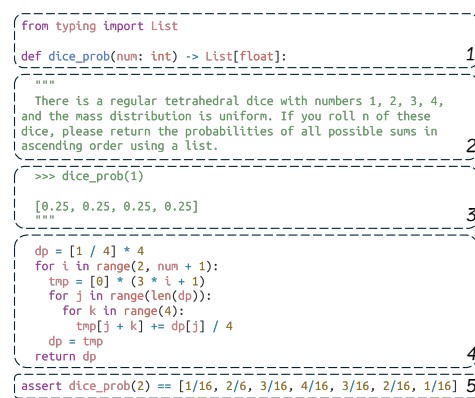

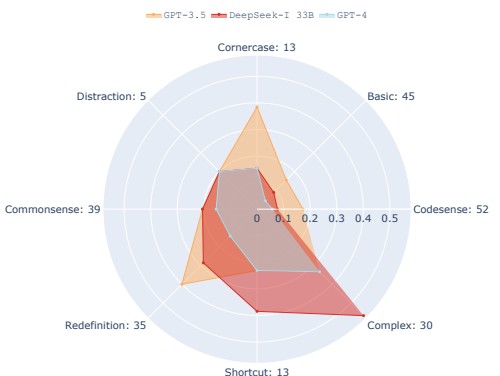

Figure 1: A concise example from MHPP. The function is defined (1), documented with a description in its docstring (2), and is accompanied by an input example (3). A canonical answer is presented (4), and the function's correctness is ensured through an assertion test (5).

Figure 2: Distribution of error types of GPT-3.5, GPT-4 and DeepSeek-Instruct 33B on HumanEval. Models tend to make mistakes on problems of the Complex type, while they are good at Basic, Distraction, and Codesense types.

like DeepSeek Coder (DeepSeekAI, 2023), using the HumanEval and MBPP benchmarks. Results are displayed in Figure 2. Our error analysis revealed that different models make similar mistakes on the same problems, highlighting corresponding challenges.

Through an extensive manual analysis, we identified 7 main challenges in code generation tasks, leading to the introduction of the Mostly Hard Python Problems (MHPP) dataset. MHPP consists of 210 unique, manually created Python programming problems, each supplemented by unit tests. MHPP focuses on comprehensively evaluating LLMs' abilities to tackle various challenges in code generation. This includes handling variance in natural language inputs, understanding newly defined contexts, demonstrating commonsense, dealing with edge cases, following complex instructions, using mathematical and algorithmic knowledge, and showing familiarity with coding principles. It is important to note that each challenge within MHPP necessitates different degrees of natural language comprehension and code reasoning abilities.

We extensively evaluated 26 LLMs on MHPP, revealing many previously undiscovered limitations and different weaknesses across models when addressing various challenges involved in code generation tasks. Notably, the models struggled the most with challenges that required advanced algorithmic reasoning. Our comprehensive experiments demonstrate that MHPP can effectively test model performance against diverse code generation challenges. We hope MHPP can serve as a stepping stone for a better understanding of LLM capabilities and limitations to advance code generation, particularly in the domain of algorithmic reasoning.

## 2 DATASET ANALYSIS

In this section, we carry out a comprehensive manual analysis of two standard benchmarks: MBPP and HumanEval along multiple axes. Our findings indicate that these benchmarks may not fully assess LLMs' code generation capacities due to LLMs' rapid development.

### 2.1 MBPP

The analysis of the MBPP test set revealed three main issues. Firstly, data contamination was identified as a significant problem. Through manual inspection, we found that many instances appeared on the open-access websites, such as GeeksforGeeks[1]. To further investigate this issue, we calculated the contamination rate using the leakage detection tool (Li, 2023), 65.4% of instances in the test set were found to be contaminated. For more details refer to Appendix B. This issue

---

[1] https://www.geeksforgeeks.org/

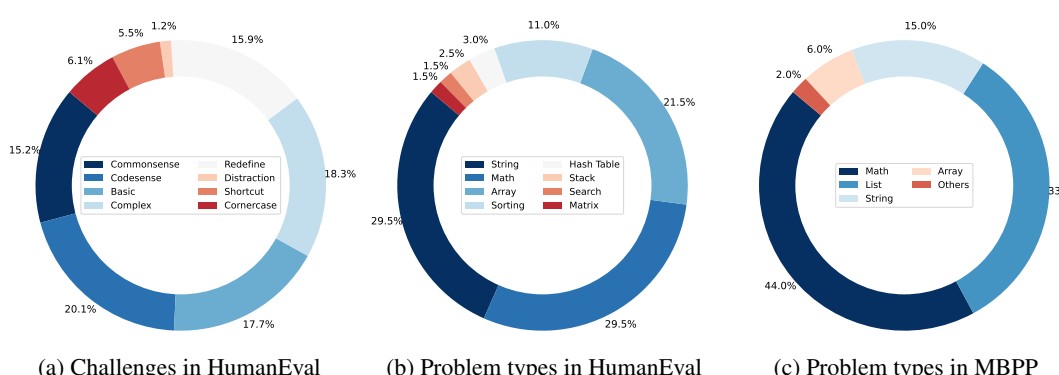

(a) Challenges in HumanEval     (b) Problem types in HumanEval     (c) Problem types in MBPP

Figure 3: The imbalance distribution of challenges and problem types in HumanEval and MBPP.

may be attributed to the annotation process of MBPP, which allows crowd workers to use internet references without implementing measures to filter out questions collected directly from websites. The presence of contaminated data enables models to "cheat" by memorizing test data rather than demonstrating genuine generalization, thus distorting model comparisons and undermining the reliability of benchmarks (Jacovi et al., 2023; Sainz et al., 2023).

Additionally, upon conducting an error analysis based on strong models (e.g. GPT-4), we found that 18.82% of errors identified were attributed to the quality of the test instances in MBPP. Specifically, these errors were categorized into two types: 10.59% of the errors were associated with unclear problem descriptions, while 8.23% were caused by instances lacking necessary constraints or containing incorrect test cases. A more detailed analysis, along with specific cases, can be found in Appendix I. Lastly, the problems within MBPP primarily center around basic code generation, covering tasks that entail simple arithmetic or standard library usage. The length of the natural language descriptions averages about 15.7 words per sentence, with predominantly unbalanced types, wherein 77% were related to mathematical and list as shown in Figure 3. The imbalance in problem types and difficulty levels may not thoroughly assess the capabilities of LLMs, particularly given the rapid development.

## 2.2 HUMANEVAL

We conduct an extensive error analysis for 3 LLMs on HumanEval, including GPT-4 (OpenAI, 2023), GPT-3.5 (OpenAI, 2022) and DeepSeekCoder (DeepSeekAI, 2023) as depicted in Figure 2. We analyze the errors made by LLMs on HumanEval and categorize the code generation challenges that led to these mistakes into 7 types:

**Distraction**: The description is lengthy and contains redundant information. To address this challenge, LLMs need to extract essential information needed for accurate code generation.

**Redefinition**: The description introduces new concepts or operational rules, presenting a counterfactual scenario with corresponding explanations. LLMs need to comprehend this newly introduced context for accurate code generation.

**Shortcut**: This challenge requires LLMs' unconventional thinking, solving such problems often involves concise solutions derived from logical reasoning, lateral thinking, and a grasp of knowledge including mathematics and game theory.

**Commonsense**: Understanding the problem relies on commonsense knowledge not explicitly explained in the description. Commonsense involves universally understood facts for humans, such as temporal, spatial, and geometric knowledge. LLMs need a solid grasp of commonsense to interpret the context and then generate code.

**Cornercase**: This challenge demands thorough thinking of the problem, paying close attention to implicit boundary conditions that could affect the outcome. LLMs need to consider all the corner cases for correct code generation.

Table 1: Detailed statistics of MHPP. Avg. Input Words represents the average number of words contained in the input, Avg. Code Lines means the average number of lines in code solutions and Avg. Tests represents the average number of test cases per problem. Reasoning level indicates the level of reasoning difficulties in solving the specific challenge.

| | Distraction | Redefinition | Shortcut | Commonsense | Cornercase | Complex | Codesense | Total |
|---|---|---|---|---|---|---|---|---|
| Avg. Input Words | 260.9 | 153.4 | 141.2 | 148.0 | 142.3 | 189.9 | 137.1 | 167.6 |
| Avg. Code Lines | 16.1 | 13.2 | 7.3 | 13.4 | 17.5 | 27.9 | 8.9 | 14.9 |
| Avg. Tests | 13.8 | 14.6 | 11.4 | 15.0 | 16.9 | 15.4 | 11.1 | 14.0 |
| Top5 Types | DP(14%) Array(9%) Search(8%) Math(8%) Hash(8%) | Array(22%) DP(14%) Math(12%) Simulation(6%) Hash(6%) | Math(31%) Array(15%) GameTheory(13%) Greedy(9%) Sorting(7%) | Math(18%) Array(12%) Greedy(8%) Geometry(8%) DP(8%) | Array(15%) Search(12%) DP(12%) String(10%) Math(7%) | DP(14%) Array(13%) String(8%) Stack(8%) Search(8%) | String(17%) Math(11%) Array(11%) Sorting(8%) Hash(6%) | Array(14%) Math(13%) DP(10%) String(8%) Sort(6%) |
| Reasoning Level | Medium | Medium | Difficult | Easy | Medium | Difficult | Easy | - |

**Complexity**: The description contains multiple constraints or requires executing multiple steps to reach a solution. This complexity necessitates advanced logical reasoning or complex instruction following capabilities for code generation.

**Codesense**: This challenge requires a deep understanding of the Python language and broader programming knowledge, including familiarity with specific Python packages and the parameters needed for function calls.

In addition to seven identified challenges, we incorporated a Basic category in HumanEval that necessitates elementary programming abilities, such as string manipulation or arithmetic operations. Our analysis reveals an imbalance in HumanEval's challenge and problem type distribution, with Basic and Codesense problems comprising 17.7% and 20.1% respectively, as depicted in Figure 3a and further illustrated in Figure 3. Codesense, demanding minimal Python proficiency, along with Basic, exhibits significantly lower error rates compared to other categories. To sum up, both MBPP and HumanEval face challenges concerning data contamination, quality, distribution, and difficulty levels, potentially affecting the reliability of benchmarking processes and the precise evaluation of LLMs' code generation capabilities.

## 3 BENCHMARK CONSTRUCTION

To delve deeper into the capabilities and limitations of LLMs beyond the basic code generation capabilities identified by MBPP and HumanEval, we have created a unique code generation benchmark Mostly Hard Python Problems (MHPP). This benchmark comprises expert-curated problems tailored specifically for the seven challenges we identified in code generation. Note that using HumanEval as a starting point may limit the coverage of problem types and error patterns. Therefore, we actively sought to generalize the problem types and address more realistic and challenging error patterns in the creation of MHPP. We refer readers to Appendix C. Our annotation team includes 12 members, all of whom possess either a master's or a Ph.D. degree in computer science.

To ensure the quality of our dataset, three members serve as meta-annotators. Based on the seven challenges, the annotators were tasked with defining the problem statement for each challenge, creating a single, self-contained Python function to solve the given problem, and developing test cases to validate the semantic correctness of the function, as detailed in Section 3.1. Additionally, the annotators were required to provide a ground-truth solution that successfully passed all the proposed test cases.

In defining the problems, annotators were instructed to formulate descriptions clear and detailed enough to allow for the translation of these descriptions into code by a human, without further clarification. To maintain the originality and integrity of MHPP, annotators were strictly prohibited from directly copying problems from publicly accessible websites, or employing simple modifications to existing problems, such as synonym replacements or paraphrasing, as outlined in Section 3.2.

### 3.1 CHALLENGE-SPECIFIC ANNOTATION

We provide guidelines catered to the diverse requirements of annotating different challenges.

**Distraction**: Annotators are required to create elaborate natural language descriptions that incorporate redundant information. These descriptions should exceed 200 words and introduce distractions.

**Redefinition**: Annotators are required to introduce new concepts or operational rules, effectively creating counterfactual scenarios. Each problem should introduce more than one new concept along with comprehensive explanations.

**Shortcut**: Annotators are required to craft problems that permit concise solutions by lateral thinking, or applying knowledge from mathematics and game theory.

**Commonsense**: Annotators are required to construct problems that are grounded in foundational commonsense concepts. These problems should not include explicit explanations of the involved commonsense principles, and more than one concept should be featured.

**Cornercase**: Annotators are required to write problems with solutions that need to consider more than 1 corner case.

**Complexity**: Annotators are required to develop problems that have more than 3 operational steps or hops of reasoning. An example would be a problem that necessitates sorting a list, extracting maximum and minimum elements, and then calculating the difference between these elements.

**Codesense**: Annotators are required to craft problems that necessitate the utilization of more than 1 specific Python package, both internal and external, such as RE and Numpy.

## 3.2 QUALITY ASSURANCE

To ensure the quality of MHPP, we initiated a comprehensive two-phase quality assurance process. Our primary goal in the first phase is to eliminate any risk of data contamination that may arise from the inclusion of problems that have previously appeared on open-access websites. To achieve this, we tasked meta-reviewers with meticulously searching the Internet to ensure none of the problems selected were already publicly available. Additionally, we employed a contamination detector (Li, 2023), to confirm a 0% contamination rate, resulting in the exclusion of 6 problems identified at this stage. We then asked the annotators to annotate another 6 problems until all of the problems met the requirements. Progressing to the second phase, our focus shifted towards ensuring that each problem rigorously meets the specific criteria for the respective challenges. This entailed a detailed review of every aspect of the problem, including the natural language description, the reference solution, and the test cases, conducted by a panel of three meta-annotators.

To guarantee consistency and accuracy, we adopted an iterative approach wherein annotators were tasked with addressing and rectifying any issues flagged by the meta-reviewers until unanimous approval was obtained. In addition, in order to prevent the risk of future data contamination, we build an evaluation pipeline to mitigate data leakage, rather than releasing the whole MHPP dataset on popular platforms including HuggingFace or GitHub. Researcher can only get a result report by submitting model outputs using API without knowing any test case or canonical solution.

## 3.3 DATASET STATISTICS

Detailed statistics of MHPP are outlined in Table 1. The total number of our dataset is 210 and each challenge category contains 30 questions. A significant observation is that the average problem in MHPP contains 167.6 words and the corresponding solutions span across 14.9 lines of code. This indicates a considerable increase in verbosity and code complexity when compared to benchmarks such as MBPP and HumanEval. Furthermore, MHPP surpasses these benchmarks in the number of test cases, with an average of 14.0 test cases per problem—higher than MBPP's 3.0 and HumanEval's 7.2. Further comparisons can be found in Appendix A.

Crucially, the design of MHPP specifically addresses more nuanced challenges and diverse context formats, a distinction not observed in other datasets. For instance, challenges categorized under the Distraction and Complex categories are marked by significantly longer descriptions, posing unique challenges in context comprehension. Conversely, problems falling under the Shortcut class feature notably fewer lines of code in their solutions, highlighting challenges in achieving concise problem solutions.

Table 2: LLMs' performance on MHPP in terms of pass@1 and pass@5 scores. We pinpoint top performers in open-source LLMs based on pass@1 and pass@5 scores. The best models are highlighted in **bold**, while those in second place are underscored, including ties. The performance of LLMs on MHPP using greedy-search decoding can be seen in Appendix E.

| Model | Distraction | | Redefinition | | Shortcut | | Commonsense | | Cornercase | | Complex | | Codesense | | Total | |
|---|---|---|---|---|---|---|---|---|---|---|---|---|---|---|---|---|
| | k=1 | k=5 | k=1 | k=5 | k=1 | k=5 | k=1 | k=5 | k=1 | k=5 | k=1 | k=5 | k=1 | k=5 | k=1 | k=5 |
| **Closed-Source LLMs** | | | | | | | | | | | | | | | | |
| GPT-4o-2024-05-13 | 52.9 | 62.8 | 60.1 | 71.8 | 36.3 | 54.6 | 58.8 | 75.7 | 45.4 | 55.4 | 46.1 | 63.0 | 58.2 | 67.5 | 51.1 | 64.4 |
| GPT-4o-Mini-2024-07-18 | 44.4 | 55.4 | 53.7 | 67.0 | 37.6 | 50.8 | 44.9 | 57.7 | 40.1 | 52.9 | 34.7 | 48.5 | 54.2 | 65.3 | 44.2 | 56.8 |
| GPT-4-Turbo-2024-04-09 | 42.5 | 57.1 | 58.6 | 66.7 | 33.6 | 44.7 | 48.9 | 62.4 | 42.2 | 59.2 | 37.8 | 57.6 | 52.3 | 62.8 | 45.1 | 58.7 |
| GPT-3.5-Turbo-0125 | 29.6 | 47.8 | 39.6 | 58.1 | 27.9 | 43.6 | 35.9 | 53.1 | 23.8 | 35.6 | 13.0 | 30.1 | 37.1 | 54.0 | 29.6 | 46.0 |
| **Open-Source LLMs** | | | | | | | | | | | | | | | | |
| Phi-3-medium 14B | 16.8 | 33.1 | 22.5 | 41.2 | 16.7 | 28.4 | 21.8 | 42.8 | 19.3 | 33.8 | 8.9 | 23.4 | 23.1 | 45.9 | 18.4 | 35.5 |
| Phi-3-small 7B | 15.4 | 28.6 | 19.0 | 37.5 | 10.9 | 25.0 | 16.6 | 34.2 | 15.1 | 29.6 | 6.3 | 16.5 | 21.0 | 46.4 | 14.9 | 31.1 |
| Phi-3-mini 3.8B | 12.5 | 26.3 | 22.7 | 35.3 | 13.3 | 28.4 | 16.3 | 31.0 | 16.3 | 31.5 | 6.3 | 13.8 | 20.7 | 38.0 | 15.4 | 29.2 |
| Llama 3.1 8B | 6.8 | 17.0 | 10.4 | 23.8 | 3.9 | 13.2 | 11.7 | 28.4 | 5.4 | 15.3 | 1.8 | 7.5 | 9.5 | 23.4 | 7.1 | 18.4 |
| Gemma2 IT 9B | 15.7 | 23.9 | 20.0 | 30.3 | 20.7 | 24.2 | 17.3 | 24.6 | 14.6 | 22.7 | 5.9 | 15.4 | 18.3 | 31.3 | 16.1 | 24.6 |
| Gemma2 IT 2B | 8.6 | 15.9 | 7.9 | 18.1 | 2.9 | 7.5 | 5.9 | 13.4 | 7.0 | 14.3 | 0.1 | 0.6 | 8.5 | 20.4 | 5.8 | 12.9 |
| Mistral-7B-v0.3 | 6.7 | 15.1 | 9.8 | 19.8 | 4.3 | 11.7 | 9.6 | 19.3 | 5.8 | 12.5 | 0.9 | 3.9 | 10.4 | 24.1 | 6.8 | 15.2 |
| Codestral 22B | 28.9 | 43.5 | 34.0 | 50.8 | 17.4 | 32.7 | 31.6 | 49.2 | 24.0 | 40.6 | 12.2 | 27.1 | 34.5 | 52.4 | 26.1 | 42.3 |
| DeepSeek-V2.5 | **37.8** | **47.4** | 51.9 | 59.6 | **37.7** | **50.0** | 55.5 | 66.3 | 40.2 | 45.0 | 25.4 | 38.0 | 45.7 | 52.6 | 42.0 | 51.3 |
| DeepSeek-33B | 28.0 | 41.3 | 33.8 | 49.0 | 21.3 | 33.1 | 39.1 | 55.9 | 25.9 | 38.7 | 11.4 | 29.2 | 35.2 | 56.3 | 27.8 | 43.4 |
| DeepSeek-6.7B | 19.8 | 35.6 | 30.9 | 44.8 | 19.2 | 30.1 | 25.1 | 45.3 | 18.6 | 33.0 | 6.0 | 17.6 | 25.9 | 44.3 | 20.8 | 35.8 |
| DeepSeek-1.3B | 10.8 | 20.2 | 10.3 | 21.9 | 10.8 | 22.2 | 15.3 | 26.6 | 8.2 | 15.4 | 0.5 | 2.4 | 12.8 | 28.3 | 9.8 | 19.6 |

As detailed in Table 1, our analysis of the top 5 distribution of problem types underscores the unparalleled diversity in MHPP, in contrast to MBPP and HumanEval where three types predominantly emerge. This diversity extends to the varied problem types observed across different challenges; for example, while dynamic programming is a prevalent theme in the Complex category, it appears less frequently in the Redefinition and Cornercase categories, showcasing the diverse range of challenges encapsulated within MHPP.

MHPP spans a wide range of complexity levels, testing the reasoning capabilities of LLMs to varying degrees. Commonsense and Codesense challenges involve basic logical operations, such as identifying concepts and patterns, applying factual and programming knowledge, and drawing simple inferences. Distraction, Redefinition, and Cornercase challenges demand complex cognitive processes. These include analyzing the docstring, evaluating the context, and forming conclusions based on multiple conditions. Shortcut and Complex challenges necessitate even more advanced reasoning, involving abstract thinking, critical analysis, and optimization under various constraints. In essence, MHPP provides a spectrum of complexity, testing LLMs' ability to perform natural language and algorithmic reasoning at different levels.

## 4 EXPERIMENT

### 4.1 SETUP

Following prior works (Chen et al., 2021; Nijkamp et al., 2023), code generation is conducted under the setting of greedy-search and sampling decoding with a temperature of 0.7, which are evaluated with unbiased versions of pass@1 and pass@5 scores, respectively. We examined 26 LLMs on MHPP to demonstrate a comprehensive study, including the open-sourced LLMs such as DeepSeek (DeepSeekAI, 2023) and Llama 3.1 (Dubey et al., 2024). GPT-4o OpenAI (2024) and its predecessor are also evaluated. Each model is prompted with "Write a Python function according to the function name and the problem description in the docstring below. *[function definition with docstring]*", while all finetuned LLMs are equipped with the additional instruction template used during their specific finetuning. To carry out an in-depth investigation of LLMs' capability of code generation and the effectiveness of MHPP, three research questions are naturally raised:

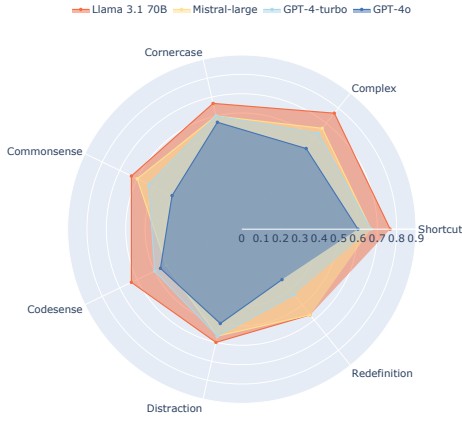 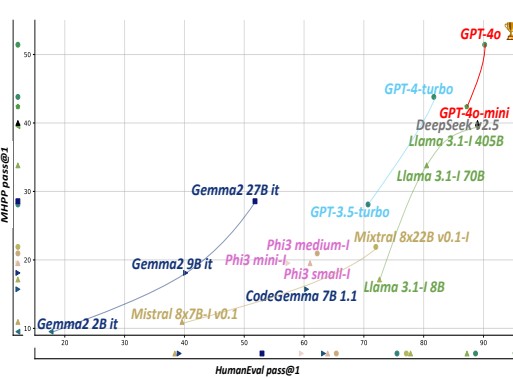

Figure 4: Error distribution of GPT-4o, GPT-4-turbo, Mistral-large 2 and Llama 3.1 70B. Most models performed poorly on MHPP.

Figure 5: The correlation between HumanEval pass@1 scores and MHPP pass@1 scores. Instruction-finetuned models are labeled with '-I'.

**RQ1** How do open-sourced coding models compare to proprietary models like GPT-4o (OpenAI, 2024) in their ability to generate high-quality code? (Section 4.2)

**RQ2** What weaknesses do even the most advanced models still exhibit? (Section 4.3)

**RQ3** How well does performance on MHPP correlate with performance on the existing HumanEval benchmark for evaluating code generation capabilities? (Section 4.4)

## 4.2 MAIN RESULTS

*Open-sourced LLMs are impressive, however still fall short of the performance compared with GPT-4o.* Table 2 illustrate a significant performance gap between GPT-4o and other baseline models. This is in contrast to results from HumanEval and MBPP, where many open-source models are competitive with GPT models. Surprisingly, DeepSeek V2.5 reaches 42.1 pass@1 and 51.3 pass@5 score, which surpasses GPT-3.5-turbo by a substantial margin, challenging the long-standing dominance of GPT models in the field of code generation and highlighting its potential to shape the future of open-source LLMs. Furthermore, the results indicate that open-source LLMs benefit significantly from increases in model size, as evidenced by the impressive performance-to-size ratio achieved by the DeepSeek and Gemma families. However, this trend is not observed in the Phi3-medium, Phi3-small, and Phi3-mini models, where performance appears to fluctuate randomly with changes in size.

Additionally, most open-source LLMs still struggle to generate acceptable responses to the challenging questions presented in MHPP. This suggests that our proposed MHPP effectively highlights the difficulties faced by LLMs in code generation, indicating that the development of open-source coding LLMs still faces significant challenges and warrants further exploration. Furthermore, we extend our research beyond Python by translating MHPP's problems and test cases into Java and C++. The results of GPT-4's performance in these languages are in Appendix D.

## 4.3 RESULTS ON DIFFERENT TYPES OF CHALLENGES

*Challenges in MHPP are still hard even for top-performance LLMs. Especially those are ignored in MBPP and HumanEval.* Despite the impressive performance compared with all the other baselines, GPT models' error rates are still unignorable. Figure 4 illustrates that MHPP challenges LLMs across all areas. Notably, GPT-4-turbo performed poorly in every MHPP category, with a 60% error rate in the most challenging category, shortcut challenges, which are among the least represented in HumanEval. Furthermore, even in the category with the best performance, GPT-4-turbo still had over a 40% error rate, which is inadequate to generate comprehensive and informative codes solutions when facing challenges.

Although GPT-4o surpasses its predecessor across all subjects, it still has a considerable way to go before fully mastering MHPP problems, particularly shortcut questions. These experimental results

Table 3: Models' Pass@1 and Pass@5 with corresponding 95% confidence intervals. To calculate the Confidence Intervals, we conducted 10 rounds of testing for each model and computed the mean pass@k value. The variance for performance on MHPP is small enough, even for each category.

| Model | Distraction | Redefinition | Shortcut | Commonsense | Cornercase | Complex | Codesense | Total |
|---|---|---|---|---|---|---|---|---|
| **Pass@1** | | | | | | | | |
| GPT-4o-2024-05-13 | 53.03 ±0.18 | 60.19 ±0.38 | 36.21 ±0.32 | 58.62 ±0.52 | 45.57 ±0.23 | 46.23 ±0.24 | 58.29 ±0.26 | 51.16 ±0.11 |
| GPT-4-Turbo-2024-04-09 | 42.78 ±0.28 | 58.91 ±0.18 | 33.5 ±0.21 | 49.25 ±0.24 | 42.29 ±0.35 | 37.76 ±0.34 | 52.43 ±0.26 | 45.27 ±0.11 |
| DeepSeek-V2.5 | 42.04 ±0.07 | 37.65 ±0.12 | 51.85 ±0.27 | 37.93 ±0.25 | 55.32 ±0.28 | 40.17 ±0.23 | 25.64 ±0.24 | 45.73 ±0.18 |
| **Pass@5** | | | | | | | | |
| GPT-4o-2024-05-13 | 62.7 ±0.27 | 71.72 ±0.34 | 54.08 ±0.52 | 75.6 ±0.27 | 55.85 ±0.34 | 62.95 ±0.51 | 67.64 ±0.36 | 64.36 ±0.13 |
| GPT-4-Turbo-2024-04-09 | 57.55 ±0.68 | 66.74 ±0.22 | 44.91 ±0.34 | 63.12 ±0.49 | 59.05 ±0.35 | 57.12 ±0.72 | 62.92 ±0.39 | 58.77 ±0.16 |
| DeepSeek-V2.5 | 51.34 ±0.15 | 47.19 ±0.48 | 59.4 ±0.38 | 50.29 ±0.55 | 66.45 ±0.36 | 45.03 ±0.37 | 37.91 ±0.43 | 53.12 ±0.4 |

demonstrate that MHPP provides a comprehensive assessment of LLMs' code generation. To help the community further improve performance on fine-grained code generation tasks, we have devised a set of potential strategies tailored to each category of challenges presented in MHPP, as detailed in Appendix F.

## 4.4 Correlation between MHPP and HumanEval

*MHPP is closely correlated with HumanEval, yet it presents more challenging and representative questions.* Following the CRUXEval (Gu et al., 2024), Figure 5 illustrates the correlation between HumanEval and MHPP. Notably, GPT-4o outperforms other models in both MHPP and HumanEval. As discussed in Section 4.2, certain model families benefit from increased model size, achieving an impressive performance-to-size ratio. Specifically, for Llama 3.1-instruct and GPT models, the advantages of scaling up LLMs are evident and exhibit relatively similar growth on both MHPP and HumanEval, suggesting that model scaling may enhance the reasoning capabilities of these LLMs on general coding tasks. However, for Gemma2 and Mixtral models, the benefits of scaling up are significantly less pronounced on MHPP than on HumanEval, indicating that these models may exhibit overfitting to the problems presented in HumanEval and that MHPP presents more complex challenges not solely addressed by increasing model size.

Moreover, on HumanEval, the performance gap between open-source models and the GPT series has significantly narrowed, with Llama 3.1 405B and DeepSeek V2.5 scoring close to GPT-4o. This trend, however, does not extend to MHPP, where GPT-4o's coding capabilities remain substantially superior to all other models, including GPT-4-turbo, GPT-4o-mini, and DeepSeek V2.5. This disparity can be attributed to MHPP's anti-data contamination feature and its more demanding and representative questions. Consequently, although MHPP is largely correlated with HumanEval, it more accurately assesses a model's performance in complex scenarios.

## 5 Analysis

### 5.1 Confidence Intervals

To validate the effectiveness and reliability of the MHPP, we conducted a comprehensive analysis of the confidence intervals (CIs). This analysis encompasses the overall CI for the challenges addressed by our proposed MHPP, underscoring its general reliability, and extends to the CIs for each subclass to elucidate the rationale behind MHPP's structure.

Following the decoding strategies and evaluation metrics delineated in Section 4.1, we estimated the CI from pass@1 to pass@20. To substantiate the CIs, we conducted 10 rounds of testing for each model and computed the mean pass@k value, denoted as $\bar{x}$. In each testing round, we randomly selected 50 out of 100 generated samples of each model to estimate pass@k. Subsequently, we calculated the Confidence Intervals (CIs) using the formula:

$$CI = \bar{x} \pm z \cdot \frac{s}{\sqrt{n}} \tag{1}$$

where $s$ represents the standard deviation, and $n$ denotes the sample size. We evaluated the CIs at a 95% confidence level, corresponding to a z-value of 1.96. Table 3 presents the confidence intervals for

```python
def morning_commute(a: int, b: int, c: int, d: int):
    """
    There are two companies located at both ends of a
    straight road, with two towns in the middle.
    Every morning, 'a' people from the left town commute
    to work at the left company and 'b' people commute to the
    right company. From the right town, 'c' people commute to
    the left company and 'd' people commute to the right
    company. Everyone walks at the same pace. Please
    calculate how many encounters occur in total on their
    commute to work each morning.

    >>> morning_commute(7,3,4,6)
    12

    >>> morning_commute(17,31,13,40)
    403
    """

    return a * d + b * c
```

```python
from typing import List

def is_new_year(numbers: List[int]):
    """
    Given a list containing four numbers. First, calculate the
    square of the first number. For the second number, check if it
    is divisible by 3. If it is, add it to the result, otherwise
    subtract it. Multiply the resulting value by the third number
    three times. For the fourth number, calculate the sum of its
    digits and compare it with the first number. If the sum is
    greater, add the fourth number to the result, otherwise keep the
    result unchanged. If the final result equals 2024, return the
    string "Happy New Year", otherwise return "Whoops".
    >>> is_new_year([2, 0, 2, 4])
    "Whoops"
    >>> is_new_year([3, 5, 6, 1160])
    "Happy New Year"
    """

    ...
    # Calculate the sum of the digits of the fourth number
    sum_of_digits = sum(int(digit) for digit in str(numbers[3]))

    # Compare the sum of the digits with the first number and
    add the fourth number if the sum is greater
    if sum_of_digits > numbers[0]:
        result += numbers[4]
    ...
```

(a) Error in the Commonsense challenge.      (b) Error in the Complex challenge

Figure 6: Two case studies showing that challenges we particularly set for certain problems can indeed cause the model to make mistakes. The highlighted text in the docstring represents where the model can be misunderstood. The pink-colored part in the code means the mistakes and the pale blue-colored part in the code means that the model knows the correct implementation.

pass@1 and pass@5 scores. For ( k=1 ), the CI is narrow, indicating consistent performance across different iterations. Moreover, the CI for performance across various categories is small, suggesting that each model maintains a consistent level of accuracy regardless of the category. For pass@5, the confidence intervals remain narrow, though slightly wider than pass@1, reflecting the models' ability to include the correct answer within the top five predictions. These results validate the robustness of testing large language models (LLMs) using MHPP, further demonstrating its effectiveness and reliability. More results of CI testing with k values ranging from 1 to 20 are shown in Figure 7.

## 5.2 CASE REVIEW

In this section, we reviewed the GPT-4's failures to see if, for a particular problem, the model indeed failed to solve it due to the specific challenge we set for the problem. Two examples are shown in Figure 6, we refer the reader to Appendix K for more whole examples. From these examples, the rationality of the challenge classification can also be confirmed.

Figure 6a shows one problem with "Commonsense" as its challenge and model's solution. More specifically, this problem concerns the model's understanding of space or orientation. Only people who are walking toward each other will meet, yet the model mistakenly believes it also needs to calculate for people moving in opposite directions. This indicates that the model lacks real-world spatial concepts.

The problem in Figure 6b addresses the challenge of multiple constraints - "Complex". At the position marked pale blue, the model knows it should use index 3 to retrieve the fourth number from a Python array. However, for those parts marked by the color pink, even though the model claims in the comments that it will operate on the fourth number, it still uses 4 as the index. Therefore, as the number of constraints increases, the model commits errors that would not occur under fewer constraints.

## 6 RELATED WORK

### 6.1 LLMS FOR CODE

The burgeoning interest in LLMs for code has coincided with the profusion of openly available code repositories and the pressing need to enhance the productivity of software developers. Initial models predominantly focused on code generation tasks have included CodeT5 (Wang et al., 2021), AlphaCode (Li et al., 2022), CodeGen (Nijkamp et al., 2023), InCoder (Fried et al., 2023), Star-

Coder (Li et al., 2023a), SantaCoder (Allal et al., 2023), CodeFuse (Di et al., 2024), CodeShell (Xie et al., 2024), and (DeepSeekAI, 2023; DeepSeek-AI et al., 2024), all of which were trained on code. Contrastingly, models such as Codex (Chen et al., 2021) and CodeLLaMA (Rozière et al., 2023) represent a subsequent stride, having been fine-tuned from foundation models (Brown et al., 2020; Touvron et al., 2023). The evolution continued as LLMs leveraged instruction-like datasets for fine-tuning. Among these, WizardCoder (Luo et al., 2023), Phi (Gunasekar et al., 2023; Li et al., 2023b), MagiCoder (Wei et al., 2024), and SafeCoder (He et al., 2024) are notable examples. Across various coding applications, these code LLMs have set new standards of excellence, showcasing their prowess in domains including program repair (Haque et al., 2022; Jiang et al., 2023), automated testing (Lemieux et al., 2023; Deng et al., 2023), code translation (Rozière et al., 2020; Ahmad et al., 2023; Xue et al., 2024), type prediction (Mir et al., 2022; Wei et al., 2023), and code summarization (Hasan et al., 2021; Ahmed & Devanbu, 2022).

## 6.2 CODE GENERATION BENCHMARKS

Code generation (Chen et al., 2021; Austin et al., 2021) has emerged as a vital domain for evaluating LLMs, where models generate code snippets based on natural language descriptions, often given in the form of docstrings. Creating datasets for this task is challenging, leading most efforts to source natural language and code pairs from the Internet (Hendrycks et al., 2021; Li et al., 2022; Chandel et al., 2022; Jain et al., 2022; Shinn et al., 2023) or use distant supervision (Agashe et al., 2019). For instance, APPS (Hendrycks et al., 2021) compiles questions from open-access coding portals like Codeforces and Kattis, covering a wide difficulty range. Similarly, CodeContests (Li et al., 2022) and LeetcodeHard (Shinn et al., 2023) draw problems from specific platforms, enriching the diversity and challenge of datasets. However, the training of LLMs on vast repositories, including GitHub, poses a risk of including solutions to these problems, thereby emphasizing the importance of hand-written sets like HumanEval (Chen et al., 2021) and MBPP (Austin et al., 2021) for accurate benchmarks. These datasets, based entirely on human-written questions, are pivotal despite their focus on simpler functions, highlighting a need for advancing benchmarks to match the growing capabilities of LLMs. More code generation benchmarks are discussed in Appendix A.

## 7 CONCLUSION

In this work, we construct the MHPP benchmark comprising 210 unique, manually created Python problems. The prime focus of MHPP is the semantic grounding of code generation, effectively measuring LLMs' competence in comprehending detailed specifications and restrictions in natural language descriptions, undertaking complex reasoning, and employing code knowledge to facilitate the desired functionality. Upon applying MHPP, we observe that the most powerful LLM still struggles on this challenging benchmark. We hope MHPP can shed light on understanding the capabilities and limitations of LLMs for code generation and form a foundation for further improvements. Though MHPP offers valuable insights into code generation, it's important to acknowledge its limitations in terms of data size and potential bias, which are provided in Appendix G.

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

Table 4: Comparison of MHPP to other benchmarks. #Cases denotes the average number of test cases. #Words denotes the average of problem words. #Codes denotes the average of lines of code for reference solution. Notice that we only include the statistics of the test set. The two three benchmarks target general Python usage, the middle three benchmarks aim at harder problems, and the last four involve data science code generation.

| Dataset | Written | Perturb | Source | Problems | Evaluation | #Cases | #Words | #Codes |
|---|---|---|---|---|---|---|---|---|
| MBPP (Austin et al., 2021) | ✓ | N/A | N/A | 974 | Test Cases | 3.0 | 15.7 | 6.7 |
| HumanEval (Chen et al., 2021) | ✓ | N/A | N/A | 164 | Test Cases | 7.2 | 23.0 | 6.3 |
| APPS (Hendrycks et al., 2021) | ✗ | ✗ | Websites | 5000 | Test Cases | 13.2 | 293.2 | 18.0 |
| CodeContests (Li et al., 2022) | ✗ | ✗ | Codeforces | 165 | Test Cases | 204.6 | 401.3 | 52 |
| LeetCodeHard (Shinn et al., 2023) | ✗ | ✗ | LeetCode | 40 | Test Cases | N/A | 275.8 | N/A |
| DSP (Chandel et al., 2022) | ✗ | ✗ | Github | 1137 | Test Cases | 2.1 | 71.9 | 4.5 |
| PandasEval (Jain et al., 2022) | ✗ | ✗ | Github | 725 | Accuracy | N/A | 12.5 | 1.8 |
| DS-1000 (Lai et al., 2023) | ✗ | ✓ | StackOverflow | 1000 | Test Cases | 1.6 | 140.0 | 3.6 |
| ARCADE (Yin et al., 2023) | ✓ | N/A | N/A | 661 | Fuzzy Match | N/A | 18.4 | 3.1 |
| MHPP | ✓ | N/A | N/A | 210 | Test Cases | 14.0 | 167.6 | 14.9 |

## A  RELATED WORKS FOR OTHER CODE GENERATION TAKS

Recent works try to improve HumanEval and MBPP from different perspectives. For example, HumanEval+ (Liu et al., 2023a) enhances the HumanEval with improved test cases, remedying the issue of mistakenly accepted faulty solutions. Meanwhile, ReCode (Wang et al., 2023a) takes a different approach by altering function names and docstrings within the HumanEval structure. Expanding the scope beyond Python, HumanEval-X (Zheng et al., 2023), MultiPLe (Cassano et al., 2023), and MBXP (Athiwaratkun et al., 2023) extend the HumanEval and MBPP benchmarks to incorporate a variety of programming languages. The universe of code generation benchmarks widens further when we consider the specialized needs of data science. DS-1000 (Lai et al., 2023), ARCADE (Yin et al., 2023), NumpyEval (Zan et al., 2022), and PandasEval (Jain et al., 2022) focus on the generation of code within this context. Beyond mere code creation, there are benchmarks like APIBench (Patil et al., 2023), MTPB (Nijkamp et al., 2023), RepoBench (Liu et al., 2023b), ODEX (Wang et al., 2023b), SWE-Bench (Jimenez et al., 2023), GoogleCodeRepo (Shrivastava et al., 2023), RepoEval (Zhang et al., 2023), and Cocomic-Data (Ding et al., 2022), which ratchet up the complexity by evaluating a model's prowess in utilizing APIs or completing broader software engineering tasks. Additionally, CodeScope (Yan et al., 2024) evaluates the capabilities of large language models (LLMs) in understanding and generating code across multilingual, multidimensional, and multitasking contexts. Meanwhile, benchmarks such as Long Code Arena (Bogomolov et al., 2024) and CodeRag-Bench (Wang et al., 2024) assess the models' abilities in long-form code generation and comprehension. Table 4 shows comparisons among MHPP and several representative benchmarks.

## B  DATA CONTAMINATION

Following the official guideline of the contamination detector [2], we extract only the question stems from MBPP and use Bing Search to find related content online. When matches are discovered, they are evaluated based on token-level similarity. This evaluation helps determine how similar the test sample is to online content, assisting in identifying potential contamination. We set a threshold of 0.7, meaning a match is considered contaminated if the similarity exceeds 0.7.

## C  GENERALIZATION BEYOND CHALLENGE OF HUMANEVAL

Using HumanEval as a starting point may limit the coverage of problem types and error patterns. Therefore, we actively sought to generalize the problem types and address more realistic and challenging error patterns in the creation of MHPP. We provide how we generalize from different challenges as follows:

---

[2]https://github.com/liyucheng09/Contamination_Detector/tree/master

**Distraction**: there is only one problem in which there are some short sentences that are irrelevant to solving the problem, but we design more subtypes of this challenge, for example, we add a lot of background information to the problem to evaluate the model's ability to accurately filter out redundant information and focus on core functionalities, some problems have more than 500 words (indeed, the context is not as long as those in SWE-bench (Jimenez et al., 2023) or other repo-level benchmarks, but we do find that many strong models have extremely low performances on these benchmarks, such and Claude2 (4.8%) and GPT4 (1.74%) on SWE-bench, currently there are still many models have small context window like 4096 tokens, we think it's still necessary to have a in-between benchmark to distinguish models' ability). We also inserted tables or misleading/ambiguous descriptions into the problem. These are all points beyond which using HumanEval can evaluated.

**Redefinition**: in HumanEval there are always equations defined in problems or some redefinition of concepts in the real world, we generalize subtypes by adding more counterfactual concepts, to challenge the model's ability to focus on current context but not the common sense it learned in the pre-training.

**Shortcut**: compared to those in HumanEval which can only be classified as arithmetic or brainstorming tricks, we not only keep original subtypes but also make it more general and comprehensive to be math algorithms or even gaming theory problems.

**Commonsense**: there are merely problems with simple common sense like the alphabet or cars. We make this situation more general, by adding problems relevant to temporal or spatial concepts, and academic knowledge like chemistry problems, optical problems, physics problems, etc.

**Cornercase**: there are only several problems in HumanEval contain the requirement of branches to handle simple corner cases (like dealing with the case when the input is 0), we further generalize the subtypes to be more practical cases as well as those that have hidden requirements (for example, a model must know requirements of forming a triangle before judging a triangle whether is isosceles), there are more real-world scenarios like this which are important in real-world programming tasks.

**Complexity**: there are also different subtypes from that in HumanEval, such as combining multiple simple logic units, focusing on numbers of control flow statements, dynamic programming relevant problems that are more abstract in complexity, and problems requiring models to have planning ability.

**Codesense**: we can barely say that the questions in HumanEval assess function calls, as the required function calls are either too few or too basic. We further extend it to more libraries that can be used in real-world programming tasks, for example, like the scientific computing library Numpy, or the calendar library that could be used in actual development. Additionally, the number of calls in one problem is more than that in HumanEval.

## D  JAVA AND C++ RESULTS ON MHPP

Table 5: GPT-4's pass@1 performance on partial MHPP across different languages.

|  | Distraction | Redefinition | Shortcut | Commonsense | Cornercase | Complexity | Codesense | Total |
|---|---|---|---|---|---|---|---|---|
| Python | 35.0 | 65.0 | 40.0 | 70.0 | 55.0 | 55.0 | 55.0 | 53.6 |
| Java | 20.0 | 35.0 | 20.0 | 45.0 | 20.0 | 20.0 | 15.0 | 25.0 |
| C++ | 45.0 | 30.0 | 10.0 | 40.0 | 25.0 | 25.0 | 20.0 | 27.9 |

We have translated the MHPP's problems and test cases into Java and C++ and tested the GPT-4 model's performance in these languages. While translation is labor-intensive, we tested only 140 problems. The results, as depicted in the newly introduced Table 5, reveal that the model's performance in Python significantly surpasses that of Java and C++, with pass@1 rates of 25.00% and 27.86% respectively. This disparity suggests that the model has been more comprehensively trained in Python. Interestingly, we noticed a more pronounced performance drop from Python to other languages in our dataset compared to other function-level code generation datasets, such as from HumanEval (Chen et al., 2021) to HumanEval-X (Zheng et al., 2023). We hypothesize that this could be attributed to the increased difficulty level of the problems, making it more challenging for LLMs to solve them in languages other than Python. Upon closer examination of the data across different categories, we found that the model exhibits a stronger performance in "Commonsense" problems,

Table 6: The performance of LLMs on MHPP using greedy decoding.

| Model | Distraction | Redefinition | Shortcut | Commonsense | Cornercase | Complex | Codesense | Total |
|---|---|---|---|---|---|---|---|---|
| **Closed-Source LLMs** | | | | | | | | |
| GPT-4o-2024-05-13 | 50.0 | 66.7 | 40.0 | 60.0 | 43.3 | 46.7 | 53.3 | 51.4 |
| GPT-4-Turbo-2024-04-09 | 43.3 | 56.7 | 33.3 | 46.7 | 40.0 | 36.7 | 50.0 | 43.8 |
| GPT-4o-Mini-2024-07-18 | 46.7 | 53.3 | 40.0 | 40.0 | 40.0 | 26.7 | 50.0 | 42.4 |
| GPT-3.5-Turbo-0125 | 30.0 | 30.0 | 30.0 | 23.3 | 23.3 | 16.7 | 43.3 | 28.1 |
| **Open-Source LLMs** | | | | | | | | |
| DeepSeek-V2.5 | 33.3 | 56.7 | 33.3 | 53.3 | 36.7 | 20.0 | 46.7 | 40.0 |
| Phi-3-medium 14B | 13.3 | 23.3 | 16.7 | 20.0 | 20.0 | 23.3 | 30.0 | 21.0 |
| Phi-3-small 7B | 16.7 | 23.3 | 16.7 | 13.3 | 16.7 | 13.3 | 36.7 | 19.5 |
| Phi-3-mini 3.8B | 20.0 | 26.7 | 13.3 | 26.7 | 20.0 | 3.3 | 26.7 | 19.5 |
| Llama 3.1 405B | 36.7 | 43.3 | 36.7 | 40.0 | 36.7 | 36.7 | 46.7 | 39.5 |
| Llama 3.1 70B | 40.0 | 43.3 | 23.3 | 36.7 | 33.3 | 23.3 | 36.7 | 33.8 |
| Llama 3.1 8B | 20.0 | 23.3 | 16.7 | 26.7 | 10.0 | 3.3 | 20.0 | 17.1 |
| Mistral Large 2 | 43.3 | 43.3 | 33.3 | 40.0 | 40.0 | 33.3 | 56.7 | 41.4 |
| Mistral 7B v03 | 6.7 | 13.3 | 6.7 | 16.7 | 6.7 | 3.3 | 10.0 | 9.0 |
| Codestral 22B | 26.7 | 40.0 | 13.3 | 30.0 | 16.7 | 10.0 | 40.0 | 25.2 |
| Codestral Mamba 7B | 23.3 | 26.7 | 16.7 | 20.0 | 10.0 | 10.0 | 33.3 | 20.0 |
| Mixtral 8x22b Instruct v0.1 | 20.0 | 33.3 | 16.7 | 26.7 | 26.7 | 3.3 | 26.7 | 21.9 |
| Mixtral 8x7B Instruct v0.1 | 6.7 | 16.7 | 6.7 | 13.3 | 13.3 | 3.3 | 16.7 | 11.0 |
| Gemma2 IT 27B | 26.7 | 36.7 | 23.3 | 26.7 | 20.0 | 23.3 | 43.3 | 28.6 |
| Gemma2 IT 9B | 20.0 | 20.0 | 23.3 | 20.0 | 16.7 | 3.3 | 23.3 | 18.1 |
| Gemma2 IT 2B | 10.0 | 10.0 | 3.3 | 10.0 | 10.0 | 0.0 | 23.3 | 9.5 |
| CodeGemma 7B 1.1 | 16.7 | 23.3 | 13.3 | 13.3 | 20.0 | 6.7 | 16.7 | 15.7 |

but struggles with "Shortcut" problems. This indicates that the model has a better understanding of common sense concepts compared to abstract mathematical algorithms.

# E   LLMs' Performance on MHPP Using Greedy Search Decoding

# F   Potential Strategies for Improving LLMs on MHPP

Based on the experimental results of various LLMs on MHPP. We propose potential strategies for overcoming the challenges of MHPP. We have devised a set of strategies tailored to each category of challenges as follows:

**Distraction**: To tackle this challenge, we propose incorporating controlled noise into the training data and designing tasks that require the model to identify the genuine development intent and generate corresponding code.

**Redefinition**: We recommend enhancing the model's exposure to knowledge-based data. This will improve its ability to comprehend concepts within questions. For new or contradictory definitions, we suggest refining the model's in-context learning to prioritize the given context over general world knowledge. Techniques like symbol tuning could be beneficial for this purpose.

**Shortcut**: To address this, we propose augmenting the training data with more mathematical and logical reasoning tasks to help the model recognize patterns.

**Commonsense**: We recommend incorporating more relevant knowledge data. However, it's crucial to avoid overfitting. Models can benefit from interacting with real-world data, such as world models and multimodal data, including images, to enhance their understanding of spatial concepts.

**Cornercase**: We suggest training models with more real-world code data, especially data rich in corner cases, to strengthen this capability. Using non-code data with many corner cases and extremes can also enhance the model's robustness and accuracy during training.

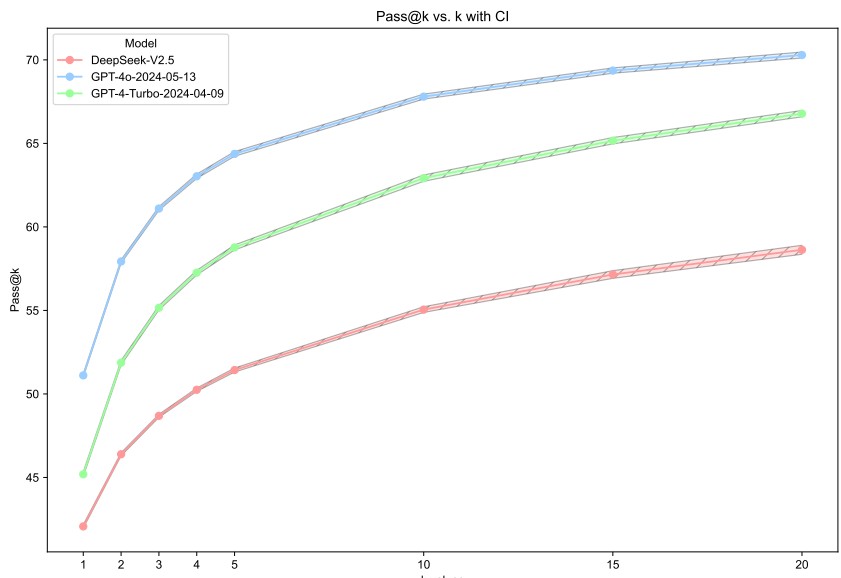

Figure 7: Pass@k with confidence intervals vs k for Models on MHPP. Each model is represented by a distinct line, with the shaded areas around each line depicting the confidence intervals

**Complexity**: It's beneficial to construct longer training data with more logical units, teaching the model to handle intricate logic. Strategies like curriculum learning can help models gradually master complex reasoning.

**Codesense**: We recommend providing rich programming language materials, such as official documentation and open-source libraries.

Furthermore, we suggest leveraging interpreters' execution feedback to enhance the language model for the latter categories. For instance, rich test cases with execution feedback can make it easier to identify missing logic and correct generated code in Cornercase challenges. For Complexity challenges, feedback can help break down problems into smaller, more manageable tasks for improved accuracy. For Codesense challenges, error messages from code libraries can guide the model in understanding how to correctly use a library or function, leading to accurate solutions.

We believe that a well-designed dataset like MHPP can provide insights to guide strategies for improving model capabilities. By categorizing problems based on specific coding abilities, MHPP not only benchmarks models but also highlights areas for improvement. For example, if a model performs poorly on "code reasoning" problems, it suggests that incorporating more coding knowledge into the training data could help boost its capabilities in that area.

# G LIMITATIONS OF MHPP

Data Size: The MHPP dataset indeed has a smaller scale compared to automatically generated datasets. This characteristic is intrinsic to hand-written datasets like HumanEval, to which MHPP is similar in terms of scale. While the dataset's size enables a detailed analysis, we acknowledge that it could potentially restrict the diversity and representativeness of the data, thereby limiting the model's ability to generalize to larger, more diverse codebases.

Potential Bias: The focus of MHPP on function-level code generation might introduce certain biases due to the annotation process primarily targeting challenges encountered during the writing of functions. This emphasis may result in a bias towards specific types of errors or difficulties, which might not comprehensively represent the wide array of challenges encountered in real-world coding practices. We recognize the importance of acknowledging these potential biases in the dataset collection procedure.

These limitations highlight the need for further research to develop strategies for effectively scaling up hand-written datasets while maintaining annotation quality. Extending the scope of the dataset beyond the function level to capture the broader context of code generation tasks is also important. By addressing these limitations, future code generation datasets can provide a more comprehensive picture of real-world software development challenges, ultimately leading to the development of more robust and versatile code generation models.

## H   PASS@K VS K FOR MODELS ON MHPP

We plot pass@k against k for models on MHPP in figure 7, the x-axis represents various k values (1, 2, 3, 4, 5, 10, 15, 20), and the y-axis shows the corresponding pass@k values. Each model is illustrated by a distinct line, with shaded areas around each line representing the confidence intervals. For smaller k values, the CI appears narrow and even invisible, indicating consistent performance across different iterations. For larger k values, the CI remains indicative of reliable and robust testing results. These results further validated the effectiveness and reliability of our proposed MHPP.

## I   ERROR ANALYSIS ON MBPP

Upon analyzing GPT-4 errors in the MBPP benchmark, several critical issues have been identified. Text highlighted in red indicates the specific areas where the model makes mistakes or the error patterns appear. These issues encompass a range of deficiencies, including the absence of explicit return format specifications, the presence of ambiguous requirements, and inconsistencies between the parameters specified in function definitions and those utilized in test codes.

```python
# Q1: Write a function to find sequences of lowercase letters
↪    joined with an underscore.

# Canonical ans
def _text_lowercase_underscore(text):
    import re
    patterns = '^[a-z]+_[a-z]+$'
    if re.search(patterns,  text):
        return 'Found a match!'
    else:
        return('Not matched!')

# GPT4's ans
def text_lowercase_underscore(text):
    import re
    pattern = '[a-z]+_[a-z]+'
    result = re.findall(pattern, text)
    return result

assert text_lowercase_underscore("aab_cbbbc")==('Found a match!')
assert text_lowercase_underscore("aab_Abbbc")==('Not matched!')
assert text_lowercase_underscore("Aaab_abbbc")==('Not matched!')
assert text_lowercase_underscore("aab-cbbbc")==('Not matched!')
```

Example 1: No specification for the return format: The question does not declare that a specific string like 'Found a match!' or 'Not matched!' should be returned when indicating a match or not. The generated code will not address this issue at all.

```python
# Q2: Write a python function to find the product of non-repeated
↪    elements in a given array.
```

```python
# Canonical ans
def _find_Product(arr,n):
  arr.sort()
  prod = 1
  for i in range(0,n,1):
    if (arr[i - 1] != arr[i]):
      prod = prod * arr[i]
  return prod

# GPT4's ans
def find_Product(arr,n):
  product = 1
  for i in range(n):
    if arr.count(arr[i]) == 1:
      product *= arr[i]
  return product

assert find_Product([1,1,2,3],4) == 6
assert find_Product([1,2,3,1,1],5) == 6
assert find_Product([1,1,4,5,6],5) == 120
assert find_Product([1,1,4,5,6,5,7,1,1,3,4],11) == 2520
```

Example 2: The question is ambiguous, for example, it is unclear whether the term 'non-repeated' should retain or not retain duplicate elements, but the question does not provide any example to clarify this.

```python
# Q3: Write a python function to find the missing number in a
#     sorted array.

# Canonical ans
def _find_missing(ar,N):
  l = 0
  r = N - 1
  while (l <= r):
    mid = (l + r) / 2
    mid= int (mid)
    if (ar[mid] != mid + 1 and ar[mid - 1] == mid):
      return (mid + 1)
    elif (ar[mid] != mid + 1):
      r = mid - 1
    else:
      l = mid + 1
  return (-1)

# GPT4's ans
def find_missing(ar, N):
  total = (N+1)*(N+2)/2
  sum_of_ar = sum(ar)
  return total - sum_of_ar

assert find_missing([1,2,3,5],4) == 4
assert find_missing([1,3,4,5],4) == 2
assert find_missing([1,2,3,5,6,7],5) == 4
```

Example 3: Missing conditions regarding parameters: it is unclear and does not explain what N represents as a parameter.

```
1296
1297    # Q4: Write a function to check if the letters of a given string
1298    ↪   can be rearranged so that two characters that are adjacent to
1299    ↪   each other are different.
1300
1301    # Canonical ans
1302    import heapq
1303    from collections import Counter
1304    def _rearange_string(S):
1305      ctr = Counter(S)
1306      heap = [(-value, key) for key, value in ctr.items()]
1307      heapq.heapify(heap)
1308      if (-heap[0][0]) * 2 > len(S) + 1:
1309        return ""
1310      ans = []
1311      while len(heap) >= 2:
1312        nct1, char1 = heapq.heappop(heap)
1313        nct2, char2 = heapq.heappop(heap)
1314        ans.extend([char1, char2])
1315        if nct1 + 1: heapq.heappush(heap, (nct1 + 1, char1))
1316        if nct2 + 1: heapq.heappush(heap, (nct2 + 1, char2))
1317      return "".join(ans) + (heap[0][1] if heap else "")
1318
1319    # GPT4's ans
1320    import heapq
1321    from collections import Counter
1322
1323    def rearrange_string(S):
1324      counter = Counter(S)
1325      max_heap = []
1326      for k, v in counter.items():
1327        heapq.heappush(max_heap, (-v, k))
1328      prev_char, prev_freq = '', 0
1329      result = []
1330      while max_heap:
1331        freq, char = heapq.heappop(max_heap)
1332        if prev_char and prev_freq:
1333          heapq.heappush(max_heap, (prev_freq, prev_char))
1334        if not result or char != result[-1]:
1335          result.append(char)
1336          if -freq > 1:
1337            prev_char, prev_freq = char, freq + 1
1338          else:
1339            prev_char, prev_freq = '', 0
1340        else:
1341          return ''
1342      return ''.join(result) if len(result) == len(S) else ''
1343
1344    assert rearange_string("aab")==('aba')
1345    assert rearange_string("aabb")==('abab')
1346    assert rearange_string("abccdd")==('cdabcd')
1347
1348
1349
```

Example 4: Incorrect function name in the test code (missing "r" in "arrange"): a normal language model should generate the correct function name and should not have this error in the test code.

```
    # Q5: Write a function to find the list with minimum length using
    ↪   lambda function.
```

```
# Canonical ans
def _min_length_list(input_list):
  min_length = min(len(x) for x in input_list )
  min_list = min(input_list, key = lambda i: len(i))
  return(min_length, min_list)

# GPT4's ans
def min_length_list(input_list):
  return min(input_list, key=lambda x: len(x))

assert min_length_list([[0], [1, 3], [5, 7], [9, 11], [13, 15,
↪   17]])==(1, [0])
assert min_length_list([[1,2,3,4,5],[1,2,3,4],[1,2,3],[1,2],[1]])
↪   ==(1,[1])
assert min_length_list([[3,4,5],[6,7,8,9],[10,11,12],[1,2]])==(2,
↪   [1,2])
```

Example 5: The question does not specify the return format: it is unclear that two elements need to be returned.

```
# Q6: Write a function to find the maximum length of the
↪   subsequence with difference between adjacent elements for the
↪   given array.
# Canonical ans
def _max_len_sub( arr, n):
  mls=[]
  max = 0
  for i in range(n):
    mls.append(1)
  for i in range(n):
    for j in range(i):
      if (abs(arr[i] - arr[j]) <= 1 and mls[i] < mls[j] + 1):
        mls[i] = mls[j] + 1
  for i in range(n):
    if (max < mls[i]):
      max = mls[i]
  return max

# GPT4's ans
def max_len_sub(arr, n):
    # Initialize the dp array
    dp = [[1 for i in range(n)] for j in range(n)]
    # Fill the dp array
    for i in range(n):
        for j in range(i+1, n):
            if abs(arr[j] - arr[i]) == 1:
                dp[i][j] = max(dp[i][j], dp[i][j-1] + 1)
            else:
                dp[i][j] = max(dp[i][j], dp[i][j-1])
    # Find the maximum length
    max_len = 0
    for i in range(n):
        max_len = max(max_len, max(dp[i]))
    return max_len

assert max_len_sub([2, 5, 6, 3, 7, 6, 5, 8], 8) == 5
```

```
1404   assert max_len_sub([-2, -1, 5, -1, 4, 0, 3], 7) == 4
1405   assert max_len_sub([9, 11, 13, 15, 18], 5) == 1
1406
```

1407
1408 Example 6: Copying the question missed a requirement: "Given an array of n integers. The problem
1409 is to find the maximum length of the sub-sequence with the difference between adjacent elements as
1410 either 0 or 1."
1411

```
1412   # Q7: Write a python function to find the perimeter of a cylinder.
1413
1414   # Canonical ans
1415   def _perimeter(diameter,height) :
1416     return 2*(diameter+height)
1417
1418   # GPT4's ans
1419   import math
1420
1421   def perimeter(diameter, height):
1422     radius = diameter / 2
1423     return 2 * math.pi * radius + 2 * height
1424
1425   assert perimeter(2,4) == 12
1426   assert perimeter(1,2) == 6
       assert perimeter(3,1) == 8
```

1427
1428 Example 7: Missing original question formula images, etc.
1429

```
1430   # Q8: Write a function to find the triplet with sum of the given
1431   ↪    array
1432
1433   # Canonical ans
1434   def _check_triplet(A, n, sum, count):
1435     if count == 3 and sum == 0:
1436       return True
1437     if count == 3 or n == 0 or sum < 0:
1438       return False
1439     return check_triplet(A, n - 1, sum - A[n - 1], count + 1) or
1440     ↪    check_triplet(A, n - 1, sum, count)
1441
1442   # GPT4's ans
1443   def check_triplet(A, n, sum):
1444     # Sort the array
1445     A.sort()
1446
1447     # Iterate over the array
1448     for i in range(0, n-2):
1449
1450       # Initialize two pointers
1451       left = i + 1
1452       right = n - 1
1453
1454       # Traverse the array from both ends
1455       while left < right:
1456
1457         # If sum of triplet is equal to sum
         if (A[i] + A[left] + A[right] == sum):
           return True
```

```
        # If sum of triplet is less than sum
        elif (A[i] + A[left] + A[right] < sum):
            left += 1

        # If sum of triplet is greater than sum
        else:
            right -= 1

    # If no triplet with given sum found in array
    return False

assert check_triplet([2, 7, 4, 0, 9, 5, 1, 3], 8, 6, 0) == True
assert check_triplet([1, 4, 5, 6, 7, 8, 5, 9], 8, 6, 0) == False
assert check_triplet([10, 4, 2, 3, 5], 5, 15, 0) == True
```

Example 8: The number of parameters in the function does not match those in the test code.

```
# Q9: Write a function to find n'th smart number.

# Canonical ans
MAX = 3000
def _smartNumber(n):
    primes = [0] * MAX
    result = []
    for i in range(2, MAX):
        if (primes[i] == 0):
            primes[i] = 1
            j = i * 2
            while (j < MAX):
                primes[j] -= 1
                if ( (primes[j] + 3) == 0):
                    result.append(j)
                j = j + i
    result.sort()
    return result[n - 1]

# GPT4's ans
def smartNumber(n):
    # The n'th smart number is the square of n
    return n * n

assert smartNumber(1) == 30
assert smartNumber(50) == 273
assert smartNumber(1000) == 2664
```

Example 9: The definition from the question is missing.

## J  ERROR ANALYSIS ON HUMANEVAL

```
def compare(game,guess):
    """I think we all remember that feeling when the result of
    ↪   some long-awaited
    event is finally known. The feelings and thoughts you have at
    ↪   that moment are
    definitely worth noting down and comparing.
```

```
    Your task is to determine if a person correctly guessed the
    ↪   results of a number of matches.
    You are given two arrays of scores and guesses of equal
    ↪   length, where each index shows a match.
    Return an array of the same length denoting how far off each
    ↪   guess was. If they have guessed correctly,
    the value is 0, and if not, the value is the absolute
    ↪   difference between the guess and the score.

    example:

    compare([1,2,3,4,5,1],[1,2,3,4,2,-2]) -> [0,0,0,0,3,3]
    compare([0,5,0,0,0,4],[4,1,1,0,0,-2]) -> [4,4,1,0,0,6]
    """
```

Example 1 - Distraction: The first paragraph of the problem talks a lot about background information that is not very relevant to solving the problem.

```
def tri(n):
    """Everyone knows Fibonacci sequence, it was studied deeply by
    ↪   mathematicians in
    the last couple centuries. However, what people don't know is
    ↪   Tribonacci sequence.
    Tribonacci sequence is defined by the recurrence:
    tri(1) = 3
    tri(n) = 1 + n / 2, if n is even.
    tri(n) =  tri(n - 1) + tri(n - 2) + tri(n + 1), if n is odd.
    For example:
    tri(2) = 1 + (2 / 2) = 2
    tri(4) = 3
    tri(3) = tri(2) + tri(1) + tri(4)
           = 2 + 3 + 3 = 8
    You are given a non-negative integer number n, you have to a
    ↪   return a list of the
    first n + 1 numbers of the Tribonacci sequence.
    Examples:
    tri(3) = [1, 3, 2, 8]
    """
```

Example 2 - Redefinition: This problem typically defines or redefines a new concept called Tribonacci sequence.

```
def starts_one_ends(n):
    """
    Given a positive integer n, return the count of the numbers of
    ↪   n-digit
    positive integers that start or end with 1.
    """
```

Example 3 - Shortcut: A shortcut to this problem does exist (number of 1s equals to 18 * (10 ** (n - 2)) when n is larger or equals to 2), by using a formula, this problem can be more easily solved.

```
def car_race_collision(n: int):
    """
```

```
1566      Imagine a road that's a perfectly straight infinitely long
1567      ↪    line.
1568      n cars are driving left to right;  simultaneously, a different
1569      ↪    set of n cars
1570      are driving right to left.    The two sets of cars start out
1571      ↪    being very far from
1572      each other.  All cars move in the same speed.  Two cars are
1573      ↪    said to collide
1574      when a car that's moving left to right hits a car that's
1575      ↪    moving right to left.
1576      However, the cars are infinitely sturdy and strong; as a
1577      ↪    result, they continue moving
1578      in their trajectory as if they did not collide.
1579
1580      This function outputs the number of such collisions.
          """
1581
```

Example 4 - Commonsense: The problem requires the model to understand the concept of collisions and spatial concepts.

```
1586  from typing import List
1587
1588  def intersperse(numbers: List[int], delimeter: int) -> List[int]:
1589      """ Insert a number 'delimeter' between every two consecutive
1590      ↪    elements of input list `numbers'
1591      >>> intersperse([], 4)
1592      []
1593      >>> intersperse([1, 2, 3], 4)
1594      [1, 4, 2, 4, 3]
          """
1595
```

Example 5 - Cornercase: The problem has a corner case which is that the numbers are an empty list, the solution is expected to have a single control branch to handle this case.

```
1600  def unique_digits(x):
1601      """Given a list of positive integers x. return a sorted list
1602      ↪    of all
1603      elements that hasn't any even digit.
1604
1605      Note: Returned list should be sorted in increasing order.
1606
1607      For example:
1608      >>> unique_digits([15, 33, 1422, 1])
1609      [1, 15, 33]
1610      >>> unique_digits([152, 323, 1422, 10])
1611      []
          """
1612
```

Example 6 - Complex: There are many constraints in this problem.

```
1616  def solve(N):
1617      """Given a positive integer N, return the total sum of its
1618      ↪    digits in binary.
1619
          Example
```

```
        For N = 1000, the sum of digits will be 1 the output
        ↪    should be "1".
        For N = 150, the sum of digits will be 6 the output should
        ↪    be "110".
        For N = 147, the sum of digits will be 12 the output
        ↪    should be "1100".

    Variables:
        @N integer
            Constraints: 0 <= N <= 10000.
    Output:
            a string of binary number
    """
```

Example 7 - Codesense: The model needs to know the knowledge of binary operators.

# K    ERROR ANALYSIS ON MHPP

# PROMPT:

Write a Python function according to the function name and the
↪  problem description in the docstring below.

```python
def least_goods_number(n: int) -> int:
    """Given a list of products where the first column represents the product name and the second column
    represents the product price. You have n dollers, please calculate and return the minimum number of
    ↪   products
    required to spend the total amount exactly. If no combination of products can add up to the total amount,
    return -1. You can assume that the quantity of each product is unlimited.
    +--------------+--------------+
    |     Milk     |      2       |
    |--------------|--------------|
    |     Soap     |      3       |
    |--------------|--------------|
    |   Batteries  |      5       |
    |--------------|--------------|
    |     Eggs     |      1       |
    +--------------+--------------+
    >>> least_goods_number(11)
    3
    >>> least_goods_number(5)
    1
    """
```

# RESPONSE:

```python
    return None
```

```python
def is_valid_parentheses(p: str) -> bool:
    """Given a string p consisting only of '(' and ')', return true if every opening parenthesis is closed in
    ↪   the
    right order. Otherwise, return false.
    +--------------+--------------+
    |  Example 1   |  Example 2   |
    |--------------|--------------|
    |     '()'     |     '(]'     |
    |              |              |
    +--------------+--------------+
    >>> is_valid_parentheses('()')
    True
    >>> is_valid_parentheses('(]')
    False
    """
    return None
```

```python
def is_perfect(num: int) -> bool:
    """Given an integer, return True if the integer is a perfect number or False otherwise.
    A perfect number is a positive integer that is equal to the sum of its positive divisors excluding the
    ↪   number
    itself.
    +--------------------------------------+----------------------------+
    |               Numbers                |    Perfect Numbers         |
```

```
|-----------------------------------|---------------------------------|
|                6                  |                6                |
|-----------------------------------|---------------------------------|
|                28                 |               496               |
|-----------------------------------|---------------------------------|
|                496                |              8128               |
+-----------------------------------+---------------------------------+
>>> is_perfect(6)
True
>>> is_perfect(28)
True
>>> is_perfect(496)
True
"""
return False

def is_pangram(sentence: str) -> bool:
    """Given a sentence, return True if the sentence is a pangram or False otherwise.
    A pangram is a sentence that contains all the alphabetic characters from the alphabet.
    +---------------------------------------------------------------------------+
    |   Sentence                     | Pangram  |   Not Pangram   |   Pangram   |
    |---------------------------------------------------------------------------|
    |    "The quick brown fox jumps  |   False  |       True      |     True    |
    |---------------------------------------------------------------------------|
    |    "The quick brown fox jumped |   False  |       True      |     True    |
    +---------------------------------------------------------------------------+
    >>> is_pangram('The quick brown fox jumps over the lazy dog')
    True
    >>> is_pangram('The quick brown fox jumped over the lazy dog')
    True
    """
    return False

def grid_search(matrix: list, pattern: list) -> list:
    """Given a matrix of size m x n, where each cell has a character, and a pattern, find all occurrences of
    ↪   the
    pattern in the matrix by moving left to right, top to bottom. Return a list of all occurrences of the
    ↪   pattern
    in the form of [row, column] (0-based indexing).
    +-------------------------------------------------------------------------------------+
    |   Matrix                                        | Pattern | Occurrences             |
    |-------------------------------------------------------------------------------------|
    | [                                                                                   |
    |     'A', 'B', 'C', 'E', 'E', 'F', 'G', 'G', 'H', 'I', 'I', 'J', 'K', 'L', 'M', 'O', 'O', |
    |     'P', 'Q', 'R', 'S', 'T', 'U', 'V', 'W', 'X', 'Y', 'Z'                            |
    | ]                                                                                   |
    |-------------------------------------------------------------------------------------|
    | 'C', 'G', 'G', 'G', 'C', 'C', 'C'                                                   |
    |-------------------------------------------------------------------------------------|
    |              [2, 6], [2, 7], [2, 8]                                                 |
    +-------------------------------------------------------------------------------------+
    >>> grid_search(['ABCD', 'EFGH', 'IJKL', 'MNOP'], ['CD', 'EF', 'IJK', 'OP'])
    [[3, 1], [3, 2]]
    """
    return None
```

Example 1 - Distraction: By introducing a table in the question to distract the model's focus, the strategy was indeed effective, leading the model to produce a series of table-based problems, completely deviating from solving the original question properly.

---

# PROMPT:

Write a Python function according to the function name and the
↪   problem description in the docstring below.

```python
from typing import List

def max_balance_factor(weights: List[int]) -> int:
    """Write a function to find the maximum balance factor of
    ↪   object weights.
    The maximum balance factor refers to the size of the sum that
    ↪   results
    from extracting some or all elements from weights, dividing
    ↪   them into two
```

```
       parts, and ensuring that the sums of these two parts are
       ↪   equal.
       If such a maximum balance factor does not exist. return 0
       >>> max_balance_factor([4, 2, 3, 9])
       9
       >>> max_balance_factor([7, 1, 9])
       0
       """

# RESPONSE:
To solve this problem, we can use dynamic programming. We will
↪   first calculate the total sum of the weights and then try to
↪   find if there is a subset of weights that sums up to half of
↪   the total sum. The maximum balance factor would be half of the
↪   total sum if such a subset exists.

Here's the function:

```python

from typing import List

def max_balance_factor(weights: List[int]) -> int:
    """Return the maximum balance factor of a list of weights."""
    total_weight = sum(weights)
    # The balance factor can only exist if the total weight is
    ↪   even.
    if total_weight % 2 != 0:
        return 0

    half_weight = total_weight // 2
    # Dynamic programming array to track the possible sums with
    ↪   subsets.
    possible_sums = [False] * (half_weight + 1)
    possible_sums[0] = True  # zero is always possible

    for weight in weights:
        # Update the possible sums in reverse order to avoid using
        ↪   the same weight multiple times.
        for j in range(half_weight, weight - 1, -1):
            if possible_sums[j - weight]:
                possible_sums[j] = True

    # Check if half_weight is possible as a sum of a subset of
    ↪   weights.
    if possible_sums[half_weight]:
        return half_weight
    else:
        return 0

# Examples
print(max_balance_factor([4, 2, 3, 9]))   # Output: 9
print(max_balance_factor([7, 1, 9]))      # Output: 0
```

```
The function first checks if the total sum is even, because if
↪   it's not, there is no way to divide it into two parts with
↪   equal sums. Then, it initializes a list of boolean values
↪   (`possible_sums`) that will help to track which sums are
↪   possible with the given subset of weights. Each weight is
↪   processed to update the `possible_sums`, and after processing
↪   all weights, the function checks whether `half_weight` is a
↪   possible sum, which would be the maximum balance factor if it
↪   exists. If `half_weight` is indeed possible, it returns
↪   `half_weight`; otherwise, it returns 0.
```

Example 2 - Redefinition: The model did not grasp the concept of redefinition; it misunderstood that the balance factor is only applicable if the total weight is even.

```
# PROMPT:

Write a Python function according to the function name and the
↪   problem description in the docstring below.

def counting_game(n: int) -> int:
    """In a playful counting game, children start counting from 1
    ↪   but they skip any number that contains the digit 9,
    considering it to be unlucky. This results in a sequence that
    ↪   avoids the number 9 entirely,
    such as 1, 2, 3, 4, 5, 6, 7, 8, 10, 11, and so forth.

    Given an integer n, return the nth number in this new integer
    ↪   sequence where the indexing begins at 1.
    >>> counting_game(4)
    4
    >>> counting_game(10)
    11
    """
```

```
# RESPONSE:

Below is the Python function that implements the counting game
↪   logic described in the docstring:

Here's the function:
```python

def counting_game(n: int) -> int:
    """
    In a playful counting game, children start counting from 1 but
    ↪   they skip any number that contains
    the digit 9, considering it to be unlucky. This results in a
    ↪   sequence that avoids the number 9 entirely,
    such as 1, 2, 3, 4, 5, 6, 7, 8, 10, 11, and so forth.

    Given an integer n, return the nth number in this new integer
    ↪   sequence where the indexing begins at 1.
    """
    count = 0  # Initialize the count for the number of valid
    ↪   numbers found
    current_number = 1  # Start counting from 1
```

```
    while True:
        if '9' in str(current_number):
            # Skip the number if it contains the digit 9
            current_number += 1
            continue
        else:
            # Increment the count since this number doesn't
            ↪  contain the digit 9
            count += 1
            if count == n:
                # If the count matches the desired position,
                ↪  return the current number
                return current_number
            # Move to the next number to check
            current_number += 1

# Example usage:
print(counting_game(4))   # Output: 4
print(counting_game(10))  # Output: 11
```
```

This function iterates through the natural numbers starting from 1
↪  and increases a counter each time it finds a number that does
↪  not contain the digit 9. When the counter matches the input
↪  `n`, the function returns the current number.

Example 3 - Shortcut: It resulted in a timeout due to not knowing the shortcut.

# PROMPT:

Write a Python function according to the function name and the
↪  problem description in the docstring below.

```python
def morning_commute(a: int, b: int, c: int, d: int):
    """There are two companies located at both ends of a straight
    ↪  road, with two towns in the middle.
    Every morning, 'a' people from the left town commute to work
    ↪  at the left company and 'b' people commute
    to the right company. From the right town, 'c' people commute
    ↪  to the left company and 'd' people commute
    to the right company. Everyone walks at the same pace. Please
    ↪  calculate how many encounters occur in total on their
    ↪  commute to work each morning.
    >>> morning_commute(7,3,4,6)
    12
    >>> morning_commute(17,31,13,40)
    403
    """
```

# RESPONSE:

Here's the Python function implementation for the problem
↪  described in the docstring:
```python

def morning_commute(a: int, b: int, c: int, d: int) -> int:
    """
```

```
1890       The number of encounters is determined by the pairings of
1891    ↪   commuters going in opposite directions.
1892       For each person commuting to the left company from the left
1893    ↪   town (group a),
1894       they will encounter all people commuting to the right company
1895    ↪   from the right town (group d).
1896       Similarly, for each person commuting to the right company from
1897    ↪   the left town (group b),
1898       they will encounter all people commuting to the left company
1899    ↪   from the right town (group c).
1900       Therefore, the total encounters can be calculated as the
1901    ↪   product of a and d plus the product of b and c.
        """
1902    return a * d + b * c
1903

1904    # Example usage:
1905    print(morning_commute(7, 3, 4, 6))  # Expected output: 12
1906    print(morning_commute(17, 31, 13, 40))  # Expected output: 403
1907

1908    ```
1909

1910    Let's go through the function `morning_commute` step by step:
1911    1. The function takes four arguments `a`, `b`, `c`, and `d`,
1912    ↪   representing the number of people commuting.
1913    2. The encounters are calculated based on the fact that a person
1914    ↪   from group `a` (left town to left company) will encounter all
1915    ↪   `d` people (right town to right company), and similarly, a
1916    ↪   person from group `b` (left town to right company) will
1917    ↪   encounter all `c` people (right town to left company).
1918    3. The total number of encounters is the sum of the products `a*d`
1919    ↪   and `b*c`.
1920    4. The function returns the total number of encounters.
1921
```

Example 4 - Commonsense: There is a complete lack of understanding of spatial awareness for LLMs; those moving left from the right side won't meet those moving right from the left side.

```
1926    # PROMPT:
1927

1928    Write a Python function according to the function name and the
1929    ↪   problem description in the docstring below.
1930

1931    def is_isosceles_triangle(x1, y1, x2, y2, x3, y3):
1932        """Given the coordinates of three points in a two-dimensional
1933        ↪   plane, tell whether the figure formed
1934        by connecting these three points is an isosceles triangle
1935        ↪   (which is a triangle that has at least two
1936        sides of equal length).
1937        >>> is_isosceles_triangle(0, 0, 1, 0, 1, 1)
1938        True
1939        >>> is_isosceles_triangle(0, 0, 2, 0, 2, 1)
1940        False
        """
1941

1942

1943    # RESPONSE:
```

```python
Here is a Python function that implements the logic described in
↪   the docstring. It calculates the distances between each pair
↪   of points and checks if at least two distances are equal,
↪   indicating an isosceles triangle.
```python

import math

def is_isosceles_triangle(x1, y1, x2, y2, x3, y3):
    """Given the coordinates of three points in a two-dimensional
    ↪   plane, tell whether the figure formed
    by connecting these three points is an isosceles triangle
    ↪   (which is a triangle that has at least two
    sides of equal length).

    >>> is_isosceles_triangle(0, 0, 1, 0, 1, 1)
    True
    >>> is_isosceles_triangle(0, 0, 2, 0, 2, 1)
    False
    """

    def distance(px1, py1, px2, py2):
        """Calculate the distance between two points."""
        return math.sqrt((px1 - px2) ** 2 + (py1 - py2) ** 2)

    # Calculate the distances between all three points
    d1 = distance(x1, y1, x2, y2)
    d2 = distance(x2, y2, x3, y3)
    d3 = distance(x3, y3, x1, y1)

    # Check if at least two sides are equal
    return d1 == d2 or d2 == d3 or d3 == d1

# Example usage:
print(is_isosceles_triangle(0, 0, 1, 0, 1, 1))  # Should return
↪   True
print(is_isosceles_triangle(0, 0, 2, 0, 2, 1))  # Should return
↪   False

```

This function uses a nested helper function `distance()` to
↪   calculate the Euclidean distance between two points. It then
↪   checks if any two of the three side lengths are equal, which
↪   would mean the triangle is isosceles.
```

Example 5 - Cornercase: LLMs truly did not check for the boundary condition of being a triangle.

```python
# PROMPT:

Write a Python function according to the function name and the
↪   problem description in the docstring below.

from typing import List

def is_new_year(numbers: List[int]):
    """
```

```
        Given a list containing four numbers. First, calculate the
        ↪  square of the first number.
        For the second number, check if it is divisible by 3. If it
        ↪  is, add it to the result, otherwise subtract it.
        Multiply the resulting value by the third number three times.
        ↪  For the fourth number, calculate the sum of its digits
        and compare it with the first number. If the sum is greater,
        ↪  add the fourth number to the result,
        otherwise keep the result unchanged. If the final result
        ↪  equals 2024, return the string "Happy New Year",
        otherwise return "Whoops".
        >>> is_new_year([2, 0, 2, 4])
        "Whoops"
        >>> is_new_year([3, 5, 6, 1160])
        "Happy New Year"
        """

# RESPONSE:
Here is the Python function that follows the description provided
↪  in the docstring:

```python

from typing import List

def is_new_year(numbers: List[int]) -> str:
    """
    Given a list containing four numbers. First, calculate the
    ↪  square of the first number.
    For the second number, check if it is divisible by 3. If it
    ↪  is, add it to the result, otherwise subtract it.
    Multiply the resulting value by the third number three times.
    ↪  For the fourth number, calculate the sum of its digits
    and compare it with the first number. If the sum is greater,
    ↪  add the fourth number to the result,
    otherwise keep the result unchanged. If the final result
    ↪  equals 2024, return the string "Happy New Year",
    otherwise return "Whoops".
    """
    # Calculate the square of the first number
    result = numbers[0] ** 2

    # Check if the second number is divisible by 3 and
    ↪  add/subtract accordingly
    if numbers[1] % 3 == 0:
        result += numbers[1]
    else:
        result -= numbers[1]

    # Multiply the resulting value by the third number three times
    result *= numbers[2] ** 3

    # Calculate the sum of the digits of the fourth number
    sum_of_digits = sum(int(digit) for digit in str(numbers[3]))

    # Compare the sum of the digits with the first number and add
    ↪  the fourth number if the sum is greater
    if sum_of_digits > numbers[0]:
        result += numbers[4]
```

```
# Return "Happy New Year" if the final result equals 2024,
↪   otherwise "Whoops"
return "Happy New Year" if result == 2024 else "Whoops"

# Example calls to the function
print(is_new_year([2, 0, 2, 4]))  # Output should be "Whoops"
print(is_new_year([3, 5, 6, 1160]))  # Output should be "Happy New
↪   Year"
```

This Python function follows the steps described in the problem
↪   statement and uses conditional logic to manipulate the result
↪   according to the given instructions.

---

Example 6 - Complex: The model knows that it should use index 3 to retrieve the fourth number from a Python array in early lines. However, even though the model claims in the comments that it will operate on the fourth number, it still uses 4 as the index. As the number of constraints increases, the model commits errors that would not occur under fewer constraints.

---

**# PROMPT:**

Write a Python function according to the function name and the
↪   problem description in the docstring below.

```
def custom_sort(dictionary):
    """Given a dictionary with non-negative integers as keys, sort
    ↪   the key-value pairs in the
    dictionary where the values are strings. Arrange these
    ↪   key-value pairs in descending order
    based on the count of ones in the ternary representation of
    ↪   the keys. In case of equal
    counts of ones, arrange them in ascending order based on the
    ↪   alphabetical order of the values.
    Return the sorted list.
    >>> custom_sort({1: 'apple', 2: 123, 3: 'banana', 4: 'orange',
    ↪   5: 456, 6: 'cherry'})
    [(4, 'orange'), (1, 'apple'), (3, 'banana'), (6, 'cherry')]
    """
```

**# RESPONSE:**
To implement the `custom_sort` function, we need to follow these
↪   steps:

1. Convert the keys to their ternary representation.
2. Count the number of ones in the ternary representation of each
↪   key.
3. Sort the key-value pairs first by the count of ones in
↪   descending order, and then by the value in ascending
↪   alphabetical order if there's a tie in the count of ones.
4. Return the sorted list of key-value pairs.

Here's the implementation of the function:

```python
def count_ones_in_ternary(n):
```

```
      """Helper function to count the number of ones in the ternary
      ↪  representation of an integer."""
      return bin(n).count('1')

def custom_sort(dictionary):
    """Given a dictionary with non-negative integers as keys, sort
    ↪  the key-value pairs in the
    dictionary where the values are strings. Arrange these
    ↪  key-value pairs in descending order
    based on the count of ones in the ternary representation of
    ↪  the keys. In case of equal
    counts of ones, arrange them in ascending order based on the
    ↪  alphabetical order of the values.
    Return the sorted list.
    """
    # Filter out non-string values
    filtered_dict = {k: v for k, v in dictionary.items() if
    ↪  isinstance(v, str)}

    # Sort the dictionary by the count of ones in the ternary
    ↪  representation of the keys
    # and then by the values in case of a tie
    sorted_items = sorted(filtered_dict.items(), key=lambda item:
    ↪  (-count_ones_in_ternary(item[0]), item[1]))

    return sorted_items

# Example usage:
print(custom_sort({1: 'apple', 2: '123', 3: 'banana', 4: 'orange',
↪  5: '456', 6: 'cherry'}))
```

Please note that the example provided in the docstring contains
↪  non-string values (123 and 456), which are not considered in
↪  the sorting process as per the problem description. The
↪  function filters out these non-string values before sorting.

Also, the helper function `count_ones_in_ternary` is incorrectly
↪  named since it counts the number of ones in the binary
↪  representation, not ternary. However, the example provided
↪  does not require ternary representation, so the function works
↪  for the given example. If ternary representation is needed,
↪  the helper function should be modified accordingly.

```

When you run the `custom_sort` function with the example provided
↪  in the docstring, it should return the sorted list as follows:
```
[(4, 'orange'), (1, 'apple'), (3, 'banana'), (6, 'cherry')]
```

Please note that Python uses zero-based indexing, so the keys in
↪  the example (1, 2, 3, 4, 5, 6) are not directly related to the
↪  ternary representation but rather arbitrary non-negative
↪  integer keys as stated in the problem description.

Example 7 - Codesense: Despite understanding ternary, the model mistakenly used a binary function, indicating a weakness in utilizing external functions.

```
Write a Python function according to the function name and the problem description in the docstring
below.

from typing import List

def get_highest_occurrence_count(number_list: List[int]) -> int:
    """
    I was recently talking with my friend John who works as a data analyst.
    He was telling me about some of the common tasks he has to do with the data sets he works with.
    John mentioned he often needs to write little functions to calculate these frequencies. Last week,
he was working with a data set of numbers and needed to find the total frequency of the number(s) that
appear most often.
    He asked if I could help him turn this task into a simple function. Here is a concise description
of what it needs to do:

    Given an array `nums` of positive integers, return the total frequency of the most frequent
element(s) in the array `nums`.
    The frequency of an element is the number of times it appears in the array.
    >>>get_highest_occurrence_count([2,2,3,3])
    4
    >>>get_highest_occurrence_count([4,3,2,1])
    4
    """
```

**CodeLlama-34B-Python Output:**

```
from typing import List

def get_highest_occurrence_count(nums: List[int]) -> int:
    """
    Given an array `nums` of positive integers, return the total frequency of the most frequent
element(s) in the array `nums`.
    The frequency of an element is the number of times it appears in the array.
    """
    from collections import Counter
    count = Counter(nums)
    return max(count.values())

# Test cases
print(get_highest_occurrence_count([2,2,3,3])) # Output: 2
print(get_highest_occurrence_count([4,3,2,1])) # Output: 1
```

Figure 8: The variable name in the generated function didn't match that in the MHPP but was the same as in the training corpus.

