# OpenReview forum: "MHPP: Exploring the Capabilities and Limitations of Language Models Beyond Basic Code Generation"
_ICLR.cc/2025/Conference — Submitted to ICLR 2025_

### Official Review · Reviewer_qDHA · 2024-11-03

**Soundness:** 3
**Presentation:** 2
**Contribution:** 2
**Rating:** 3
**Confidence:** 4

**Summary:**

This paper introduces MHPP, a new benchmark for code generation that seeks to address the limitations of existing benchmarks like HumanEval and MBPP, which have become too simple for current high-performing language models such as GPT-4. MHPP contains 210 human-curated Python problems, designed in a similar code completion style to HumanEval but with enhanced diversity and difficulty across various problem types (categorized as 7 challenges). Evaluation results on 26 LLMs reveal that models performing well on HumanEval do not achieve comparable results on MHPP, highlighting the increased complexity of this benchmark. Additional analysis also examines the types of errors made by models like GPT-4 and confirms a correlation in performance trends with HumanEval.

**Strengths:**

Overall, it is a good attempt to address the limitations of existing code generation benchmarks with more finegrained and complex evalution. Extensive evaluation show that this increase in complexity helps reveal the performance limits of advanced LLMs, which perform well on simpler benchmarks but struggle with MHPP.

**Weaknesses:**

- **Incremental Contribution**: While this paper makes a commendable attempt to address the limitations of prior code generation benchmarks by offering a more fine-grained evaluation, its novelty and technical contributions are relatively incremental. It aligns with a line of research that introduces increasingly challenging benchmarks to highlight the limitations of LLMs without substantially advancing our understanding of how to overcome these challenges. Additionally, the insights drawn from the results are somewhat **limited**, especially considering that other more complex code generation benchmarks, such as LiveCodeBench, already exist. To make it more solid work, the authors should provide more insights from the results and how they guige targeted improvements to model architectures or training. For example, they can analyze patterns in the errors to gain insights into fundamental limitations of current LLM approaches.
- **Unsystematic Problem Curation**: The problem categories and challenges are curated manually, which may affect consistency, coverage, and diversity. The lack of an automated, systematic process for problem generation could limit the scalability and reproducibility of this benchmark, making it harder to ensure high quality and comprehensive coverage across problem types.

**Questions:**

* How to ensure high diversity and comprehensive coverage of fully human-crafted benchmarks?
* Could you provide a summary of the key insights derived from the experimental results?
* Are there any potential approaches to enhance LLMs to better handle these challenges uncovered by MHPP?

---

> ### Author Response · Authors · 2024-11-28
>
> 1. Contribution and novelty
>
> Thank you for your constructive feedback. We agree that our benchmark offers a more fine-grained evaluation, which provides valuable insights into the strengths and weaknesses of LLMs. As we mention in Appendix J, we have indeed conducted further analysis on error patterns to highlight model performance issues across different categories. We believe this analysis helps us identify specific weaknesses in model performance that would be hard to detect using more general benchmarks. Additionally, we also discuss potential strategies for improving LLMs in Appendix F, where we explore targeted improvements based on the errors observed in specific challenge categories.
>
> Regarding the novelty, we believe that the fine-grained categorization of problems is one of the main contributions of our work. It allows us to focus on particular problem types, which helps guide more targeted improvements without the need for extensive error case analysis. This approach saves significant human effort, as analyzing every individual error case in large-scale code datasets would be highly time-consuming and inefficient.
>
> We understand the concern about the contribution being perceived as incremental, but we believe that this granularity of analysis and the potential for actionable improvements in model architectures make it a meaningful addition to existing research.
>
> 2. Automated generation could lead to biases
>
> Thank you for raising this point. While we acknowledge that our benchmark relies on manual curation, we intentionally chose this approach over automated problem generation. Automated generation could lead to significant biases, as the generated problems are often ones that models are already good at solving. This would limit the diversity and real-world applicability of the problems. Furthermore, if automated systems were to generate problems, they could potentially introduce a bias where certain models that have been trained on similar datasets might perform disproportionately well on those problems.
>
> In contrast, our manual curation process ensures a more careful and thoughtful selection of problems. We have statistics demonstrating the diversity of the problems we include, as we examine a wide range of problem types. Additionally, we employ meta annotators to ensure that the problems do not overlap or duplicate, which sets our dataset apart from others, such as HumanEval, where there are even highly similar problems. While comprehensive coverage of all possible problem types may not be feasible, we have designed our benchmark with different categories to ensure broad coverage. We do not believe that a "perfect" dataset can cover all possible programming challenges, but we aim to cover a representative variety of problem types to test model performance across different aspects of code generation.
>
> 3. Curation and evaluation pipelines ensure diversity and reproducibility
>
> Our approach prioritizes high-quality, human-curated problems to ensure that the problems are challenging and not trivially solvable by current models. We follow a reliable process during dataset construction, starting with analyzing existing datasets and identifying key challenges. After generating problems, we refine and adjust them multiple times to ensure their quality. This manual curation process helps us create diverse, robust problem categories that represent a wide range of coding challenges.
>
> Regarding reproducibility, we have established a clear evaluation pipeline that guarantees reproducibility of results, and the benchmark has been in use for an extended period with consistent results. This pipeline has been tested and verified, ensuring that future use of the benchmark will provide consistent and reliable evaluations.
>
> In summary, we believe that manual curation is essential to prevent bias and overfitting to models, and our well-established, reproducible process ensures the quality and diversity of the dataset.

---

### Official Review · Reviewer_daQc · 2024-11-03

**Soundness:** 2
**Presentation:** 3
**Contribution:** 2
**Rating:** 3
**Confidence:** 4

**Summary:**

The paper proposes a new coding benchmark that is intended to be more challenging for Code LMs than MBPP and HumanEval. It also undertakes an analysis of the standard failure modes of models on existing benchmarks as well as leakages of existing benchmarks in the pre-training data.

**Strengths:**

1. The benchmark is human-created and (for the moment) is unlikely to be a part of any pre-training corpora
2. The authors show that the problems are challenging enough to leave some headroom, even for the SOTA models

**Weaknesses:**

Overall, I do not see the point of this benchmark in terms of bringing something to the field that is not already out there:

1. ~14 tests on average per sample makes it better than HumanEval and MBPP but is still far outmatched by benchmarks such as EvalPlus [1]
2. In terms of being a challenging test for CodeLMs, due to limitations in chosen domains, library usage and question difficulty, it is, on average, well outdone by existing benchmarks like BigCodeBench [2], ClassEval [3] and SWE-Bench [4].
3. It is also a Python-native benchmark, which leaves out medium and low-resource programming languages that existing benchmarks like McEval [5], Multipl-e [6] or MxEval [7] cover.

[1] https://arxiv.org/abs/2305.01210
[2] https://arxiv.org/abs/2406.15877
[3] https://arxiv.org/abs/2308.01861
[4] https://arxiv.org/abs/2310.06770
[5] https://arxiv.org/abs/2406.07436
[6] https://arxiv.org/abs/2208.08227
[7] https://arxiv.org/abs/2406.07436

**Questions:**

It would be very useful for the authors to more explicitly situate this benchmark in terms of existing work and what it uniquely brings to the table.

---

> ### Author Response · Authors · 2024-11-28
>
> 1. Contribution and Novelty
>
> Thank you for your comment.
>
> * One of the main contributions of our work is the introduction of many challenging, clean code problems, which differ from those in other benchmarks that are mostly derived from publicly available data. The new problems we include require significant manual effort to curate, and we have also established a robust evaluation pipeline for testing. Unlike other datasets, where answers can be easily found, our approach minimizes the risk of data contamination, which is crucial for ensuring the integrity of future evaluations.
>
> * Another primary contribution of our work lies in the more detailed categorization of problem types, which enables us to observe how models perform across a range of challenges. This fine-grained classification helps identify performance gaps, which can guide targeted improvements in models without the need for extensive error case analysis. Unlike existing benchmarks, which typically provide broad categories, our approach allows for deeper insights into specific areas of model weakness and the potential for improvement.
>
> 2. Number of tests
>
> We acknowledge that EvalPlus has more tests per problem. However, we believe that as long as coverage is sufficient to test core model capabilities, additional tests may add redundancy. Our coverage tests confirm that we strike a balance by keeping a smaller but diverse set of problems, reducing unnecessary repetition. Additionally, referring to the ClassEval benchmark, we observed from Table 1 in its paper that our manually curated dataset already contains a sufficiently large number of test cases compared to other manually created code datasets. This supports our approach of prioritizing quality over sheer quantity.
>
> 3. Challenging for CodeLMs and Domain Limitations
>
> We understand your concerns, but we believe that our benchmark is addressing a complementary need. While BigCodeBench, ClassEval, and SWE-Bench focus on different aspects (e.g., utilizing function calls as tools, generating a class of multiple interdependent methods or software engineering issues), our dataset emphasizes code reasoning and logical problem-solving. This aspect is currently underrepresented in existing benchmarks, making our contribution distinct and necessary.
>
> For example, our Shortcut category tests models' ability to apply concepts like game theory, which is not a focus of the other benchmarks. This provides a new dimension to model evaluation that has been largely absent from existing datasets.
>
> 4. Python-Specific Nature of the Benchmark
>
> Thank you for your comment. While many of the problems in our benchmark are implemented in Python, we want to emphasize that the majority of the tasks are not Python-specific; they focus on algorithmic concepts and logical problem-solving, which can be translated to other programming languages.
>
> For example, as shown in the Appendix D of the paper, we have already translated 140 problems into different languages. Additionally, during the rebuttal period, we completed a full translation of the entire 210-problem dataset into C++. The performance of o1-mini on this C++ dataset is shown in Rebuttal Table 3. While performance remains suboptimal across languages, this demonstrates the flexibility of our benchmark to be applied to non-Python languages as well.
>
> We will further explore and expand the benchmark to include more language versions in the future.
> | Model   | Distraction | Redefination | Shortcut | Commonsense | Cornercase | Complex | Codesense | Total |
> |---------|-------------|--------------|----------|-------------|------------|---------|-----------|-------|
> | o1-mini | 33.3        | 13.3         | 33.3     | 33.3        | 20         | 16.7    | 20        | 24.3  |
>
> Rebuttal Table 3: Results for o1-mini on MHCP Dataset

---

> ### Comment · Reviewer_daQc · 2024-12-02
> **Thank You for the Response**
>
> While I acknowledge the upgrade this benchmark represents in terms of difficulty viz-a-viz existing ones like HumanEval, I still think this benchmark doesn't bring anything new compared to tough benchmarks like BigCodeBench-Hard which have been shown to be challenging and also have low overlap with pre-training corpora. With this in view, I have decided to retain my scores.

---

### Official Review · Reviewer_YQL9 · 2024-11-03

**Soundness:** 3
**Presentation:** 4
**Contribution:** 2
**Rating:** 5
**Confidence:** 5

**Summary:**

The paper introduces a new dataset called Mostly Hard Python Problems (MHPP), consisting of 210 unique, human-curated Python programming problems designed to evaluate LLMs' abilities across seven manually identified challenges. Next, it presents experiments on 26 LLMs providing empirical insights.

**Strengths:**

* **Benchmark and Error Analysis.** The authors analyze existing benchmarks MBPP and HumanEval identifying mistakes and contamination. They also produce a manual categorization of mistakes made on HumanEval
* **New benchmark.** The authors provide a manually curated benchmark of 210 problems focusing on mistakes identified on
HumanEval. This provides confidence in the quality of the benchmark problem statements
* **Qualitative Analysis.** The authors provide insights into model failures through detailed qualitative analysis on existing and proposed benchmarks.

**Weaknesses:**

* **Benchmark Size.** The benchmark only consists of 210 benchmark problems. The benchmark size is limited and puts concerns over empirical findings in question. This is an even more serious concern for problem for the category-wise results presented where the number of benchmark samples would be around 30-40 (Tables 2, 3 and Figure 4) -- potentially increasing noise levels to over 10/15% making the results unreliable.

* **Confidence Intervals.** As I understand, section 5.1 computes the confidence intervals for noise introduced due to temperature sampling. This should be properly clarified since it does not account for variability due to the limited size of the benchmark which is a considerably bigger source of noise. This is also a known issue with HumanEval (see [1]).

* **Related Work.** The paper does not clarify or distinguish from many programming benchmarks released in the past year including but not limited to xCodeEval, CodeScope, RepoEval, RepoBench, ClassEval, LiveCodeBench, EvoEval, BigCodeBench.

MHPP is particularly similar to EvoEval [2] which identifies overlapping sets of problem types `Difficult, Creative, Subtle, Combine, and Tool Use` against `Redefinition, Distraction, Shortcut, CommonSense, Cornercase, Complexity and Codesense` studied in this work. [2] additionally proposed an approach to automatically curate problems from existing benchmarks -- however at the cost of not providing sufficient guarantees on problems - a strength of manual curation performed by the authors. The related works should be discussed and differences clarified appropriately

* **Novelty.** While collected through manual curation, the paper does not challenge existing empirical performance trends (beyond the reduced performance on the more challenging benchmark). However, this should not diminish the usefulness of the benchmark problems considering saturation in HumanEval.

[1] https://github.com/crux-eval/eval-arena

[2] Top Leaderboard Ranking = Top Coding Proficiency, Always? EvoEval: Evolving Coding Benchmarks via LLM. COLM 2024

**Questions:**

* Can the authors clarify the confidence intervals constructed differentiating the types of noise observed? Additionally, it might be useful to measure the noise in evaluating the performance of 30/40 samples using a bootstrapping-based approach (sample n problems out of 210 and measuring noise).

---

> ### Author Response · Authors · 2024-11-28
>
> 1. Benchmark Size and Confidence Intervals
>
> We acknowledge the concern regarding the limited size of the benchmark. To address this, we re-examined the noise levels using a bootstrapping-based approach, similar to methods employed in Eval-Arena and CruxEval. Specifically, we sampled 25 problems from each category (totaling 175 problems) and performed 1000 iterations to estimate the variability across the dataset. The results for GPT-4o, shown in Rebuttal Table 2, indicate that for the entire dataset, the noise levels are manageable. However, for certain categories, the noise approaches 10%, which may impact the reliability of results.
>
> To mitigate this issue in future work, we plan to expand the dataset to include over 1,000 problems and apply Eval-Arena's methodology for evaluating noise more rigorously. This will help reduce noise and improve the robustness of our results.
>
> We also appreciate the reviewer’s observation about the confidence interval analysis in Section 5.1. While the current analysis primarily addresses noise introduced by temperature sampling, it does not fully account for variability due to the benchmark size, which we now acknowledge as a larger source of noise.
>
> | Model  | Distraction | Redefination | Shortcut | Commonsense | Cornercase | Complex | Codesense | Total |
> |--------|-------------|--------------|----------|-------------|------------|---------|-----------|-------|
> | gpt4o  | 2.94        | 8.23         | 5.87     | 9.99        | 6.5        | 9.56    | 8.58      | 7.68  |
>
> Rebuttal Table 2: Noise Levels for GPT-4o with Bootstrapping Sampling.
>
> 2. Comparison to EvoEval and Related Work
>
> Thank you for pointing out the similarities to EvoEval. While there are some overlapping categories (e.g., Tool Use in EvoEval vs. Codesense in our work), the two benchmarks differ significantly in scope and objectives. Our Codesense tests a model's ability to call external functions and libraries, while EvoEval’s Tool Use focuses on function decomposition, which is a simpler task (we provide a comparison of examples below.).
>
> ```
> #example in EvoEval
> def check_vowel(s):
>    """helper function"""
>
> def frequency_count(s):
>    """Given a string s, count the frequency of each vowel in the string. Return the results as a dictionary. """
>
> #example in MHPP
> def get_specific_permutation(N: int, M: int) -> str:
>    """Given a integer N (1 <= N <= 9), there are n! different permutations that can be formed using digits less than or equal to N. If we arrange these permutations in ascending order, for example, when n=3, the 1st result is 123, and the 6th result is 321. Please find the Mth permutation when given the values of N and the positive integer M.
>    >>> get_specific_permutation(3, 6)
>    "321"
>    >>> get_specific_permutation(4, 9)
>    "2314"
>    """
> ```
>
> More importantly, EvoEval does not introduce new problem categories, unlike our work, which creates novel categories such as Shortcut, designed to test advanced knowledge like game theory—something EvoEval struggles to address. This distinction allows us to better evaluate models on a wider range of abilities.
>
> We also have concerns that automated generation could introduce significant biases, as the problems generated are often ones that models are already adept at solving. This would limit the diversity and real-world applicability of the problems. Additionally, if automated systems were used to generate problems, they could introduce a bias where models trained on similar datasets might perform disproportionately well on those specific problems.
>
> We will expand on these differences in the related work section and plan to conduct a detailed comparison in future revisions. We also believe that combining EvoEval’s problem-generation approach with our dataset could further enhance our benchmark.
>
> 3. Novelty and Contribution
>
> Thank you for your feedback.
>
> * One of the main contributions of our work is the introduction of many challenging, clean code problems, which differ from those in other benchmarks that are mostly derived from publicly available data or generated by models. The new problems we include require significant manual effort to curate, and we have also established a robust evaluation pipeline for testing. Unlike other datasets, where answers can be easily found, our approach minimizes the risk of data contamination, which is crucial for ensuring the integrity of future evaluations.
> * The main novelty of our work also lies in the more detailed categorization of problems, which allows us to observe how models perform across different types of challenges. This granularity helps identify performance gaps, guiding targeted improvements without the need for extensive error case analysis.
>
> Although our benchmark does not directly challenge existing performance trends, it offers valuable insights, especially in light of the saturation in benchmarks like HumanEval. We believe this more nuanced approach will help drive future model improvements.

---

> > ### Comment · Reviewer_YQL9 · 2024-12-03
> >
> > > ... for the entire dataset, the noise levels are manageable. However, for certain categories, the noise approaches 10%, which may impact the reliability of results.
> >
> > Thank you for computing the noise levels. I strongly encourage the authors to add this analysis to the paper and supplement the benchmark with additional problems as possible. Given the higher error margins, however, I believe the existing results might need to be reinterpreted and I prefer to keep my current score.

---

### Official Review · Reviewer_whjS · 2024-11-04

**Soundness:** 3
**Presentation:** 3
**Contribution:** 3
**Rating:** 6
**Confidence:** 3

**Summary:**

This paper presents a function-level code generation benchmark comprising 7 distinct problem types. With state-of-the-art models like GPT-4o achieving a Pass@1 of 51.1, this benchmark offers a valuable resource for advancing future work in the community.

**Strengths:**

- A key question about whether current LLMs have mastered function-level code generation, and the detailed breakdown of 7 challenge types effectively motivate the need for this benchmark.

**Weaknesses:**

- The benchmark includes 210 problems, which, while comparable to HumanEval’s 164, may be insufficient for broader generalizability. Note that recent benchmarks, like BigCodeBench [1], offer over 1K problems.

- Current code generation benchmarks including this work are vulnerable to future data contamination as the test set is often public. To mitigate this, splitting the benchmark into validation and hidden test sets, with evaluations on the test set limited to submissions, may be advisable, following NL2SQL [2, 3].

[1] Zhuo, Terry Yue, et al. "Bigcodebench: Benchmarking code generation with diverse function calls and complex instructions." arXiv preprint arXiv:2406.15877 (2024).

[2] Tao Yu, Rui Zhang, Kai Yang, Michihiro Yasunaga, Dongxu Wang, Zifan Li, James Ma, Irene Li, Qingning Yao, Shanelle Roman, Zilin Zhang, and Dragomir Radev. 2018. Spider: A Large-Scale Human-Labeled Dataset for Complex and Cross-Domain Semantic Parsing and Text-to-SQL Task. In Proceedings of the 2018 Conference on Empirical Methods in Natural Language Processing, pages 3911–3921, Brussels, Belgium. Association for Computational Linguistics.

[3] Jinyang Li, Binyuan Hui, GE QU, Jiaxi Yang, Binhua Li, Bowen Li, Bailin Wang, Bowen Qin, Ruiying Geng, Nan Huo, Xuanhe Zhou, Chenhao Ma, Guoliang Li, Kevin Chang, Fei Huang, Reynold Cheng, & Yongbin Li (2023). Can LLM Already Serve as A Database Interface? A BIg Bench for Large-Scale Database Grounded Text-to-SQLs. In Thirty-seventh Conference on Neural Information Processing Systems Datasets and Benchmarks Track.

**Questions:**

- While not essential, including results for GPT-o1-preview could further underscore the point that current LLMs have yet to fully master function-level code generation, particularly if its performance is lower.

---

> ### Author Response · Authors · 2024-11-28
>
> 1. Benchmark Size and Comparisons to BigCodeBench
>
> We appreciate the reviewer pointing out the comparison to BigCodeBench. However, we emphasize that BigCodeBench is fundamentally different from our benchmark in terms of design and focus. We provide a comparison of examples below. BigCodeBench is not artificially constructed and centers on real-world scenarios without algorithmic problems, which aligns with a different category of problem types. In contrast, our benchmark focuses specifically on algorithmic problem-solving, targeting a direct comparison with HumanEval. This deliberate focus allows us to evaluate algorithmic reasoning capabilities more effectively.
>
> Furthermore, our benchmark is artificially constructed, which reduces the likelihood of data leakage—a common challenge in real-world datasets. Additionally, this approach minimizes biases in problem distribution, ensuring a balanced evaluation of algorithmic understanding. While the size of the dataset may appear smaller than BigCodeBench, we conducted extensive variance analysis to demonstrate that the dataset size is sufficient to ensure reliable and stable performance evaluation.
>
> ```python
> # example in BigCodeBench
> import http.client
> import socket
> import ssl
>
> def task_func(server_name, server_port, path):
>    """
>    Makes an HTTPS GET request to a specified
>    server and path, and retrieves the response.
>    Parameters: ...
>    Returns: ...
>    Raises:...
>    Requirements:...
>    Examples:...
>    """
>
> # example in MHPP
> from typing import List
>
> def generate_string_permutation(raw_data: List[str], replaced_str: str) -> int:
>    """Generate all permutations of raw_data elements, excluding one element per permutation.
>    Replace up to three occurrences of replaced_str in each permutation with an empty string.
>    Return the number of unique strings.
>    >>> generate_string_permutation(["a", "b", "c"], "a")
>    4
>    >>> generate_string_permutation(["hello", "world", "rock", "great"], "or")
>    24
>    """
> ```
>
> 2. Data Contamination Concerns
>
> We understand the concern about data contamination and have taken significant steps to mitigate this issue in our benchmark. Specifically:
> * Limited Test Set Exposure: Only a minimal portion of the test set (3 test cases for each problem) has been made publicly available, ensuring the vast majority of test problems remain hidden.
> * Evaluation Process: We have designed an evaluation pipeline where participants can only get a result report by submitting model outputs using API without knowing any further test case or canonical solution.
>
> We are confident that this pipeline effectively mitigates data contamination risks, as our problems were made available on GitHub early on, and after applying the contamination detector mentioned in Section 3.2, the current contamination rate remains 0%.
>
> 3. Additional Results for o1-preview
>
> Thank you for the suggestion. We agree that including o1-preview results strengthens our argument. As shown in Rebuttal Table 1, the evaluation results confirm that o1-preview performs lower than other benchmarks (e.g., 89.0 on Codeforces), highlighting the ongoing challenges faced by current LLMs in mastering function-level code generation. We will include these results in the revised paper to further emphasize this point.
>
> | Model            | distraction | redefination | shortcut | commonsense | cornercase | complex | codesense | total |
> |------------------|--------------------|---------------------|-----------------|--------------------|-------------------|----------------|------------------|--------|
> | o1-preview| 80                 | 66.7                | 70              | 70                 | 53.3              | 63.3           | 73.3             | 68.1   |
> | o1-mini          | 70                 | 70                  | 76.7            | 66.7               | 63.3              | 50             | 66.7             | 66.2   |
>
> Rebuttal Tabel 1: o1-preview and o1-mini performance on MHPP

---

### Meta-Review · Area_Chair_7qMu · 2024-12-16

**Metareview:**

The paper introduces MHPP, a dataset of 210 unique, human-curated Python programming challenges aimed at assessing LLM performance across seven challenge types. These challenges types are inspired by manual inspections from existing benchmarks HumanEval and MBPP. The work reports experimental results from 26 LLMs, demonstrating the challeges of MHPP.

While all reviewers agreed the dataset can be valuable, the work lacks a convincing statement of its novelty. Particularly (1) the comparison with many existing benchmarks such as EvolEval, CruxEval is unclear (2) The dataset size is small, and the data collection process relies heavily on human efforts, leaving scaling the dataset size a challenge.

**Additional Comments On Reviewer Discussion:**

NA

---

### Decision · Program_Chairs · 2025-01-22

Reject